

# Evaluating the Potential of PPK Direct Georeferencing for UAV-SfM Photogrammetry and Precise Topographic Mapping

He Zhang[1], Emilien Aldana-Jague[1], François Clapuyt[1], Florian Wilken[2], Veerle Vanacker[1], and Kristof Van Oost[1]

[1]Earth and Life Institute, Georges Lemaître Centre for Earth and Climate Research, Université Catholique de Louvain, Louvain-la-Neuve, 1348, Belgium
[2]Institute for Geography, Universität Augsburg, Augsburg, 86159, Germany

*Correspondence to*: He Zhang (he.zhang@uclouvain.be)

**Abstract.** Images captured by Unmanned aerial vehicle (UAV) and processed by Structure from Motion (SfM) photogrammetry are increasingly used in geomorphology to obtain high resolution topography data. Conventional georeferencing using ground control points (GCPs) provides reliable positioning but the geometrical accuracy critically depends on the number and spatial layout of the GCPs. This limits the time- and cost-effectiveness. Direct georeferencing of the UAV images with differential GNSS, such as PPK (Post-Processing Kinematic), may overcome these limitations by providing accurate and directly georeferenced surveys. To investigate the positional accuracy, repeatability and reproducibility of digital surface models (DSMs) generated by a UAV-PPK-SfM workflow, we carried out multiple flight missions with different camera/UAV systems. Our analysis showed that the PPK solution provides the same accuracy (mean: ca. 0.01 m, RMSE: ca. 0.03 m) as the GCP method. Furthermore, our results indicated that camera properties (i.e., focal length, resolution, sensor quality) have an impact on the accuracy but planimetric and altimetric errors remained in the range of 0.011 to 0.024 m. By analysing the repeatability of DSM construction over a time period of a few months, our study demonstrates that a UAV-PPK-SfM workflow can provide consistent, repeatable 4D data with an accuracy of a few centimetres without the use of GCPs. An uncertainty analysis showed that the minimum level of topographical change detection was ca. ±0.04 m for a high-end DSLR camera and ca. ±0.08 m for an action camera (for a flight height of 45 m). The level of detection substantially improved when reducing the UAV flight height. This study demonstrates the repeatability, reproducibility and efficiency of a PPK-SfM workflow in the context of 4D earth surface monitoring with time-laps SfM photogrammetry. As such, it should be considered as an efficient tool to monitor geomorphic processes accurately and quickly at a very high spatial and temporal resolution.

## 1 Introduction

During the past decade, Unmanned Aerial Vehicles (UAVs) or Unmanned Aerial Systems (UAS) are increasingly used in geomorphological research (Passalacqua et al., 2015; Tarolli, 2014). They are currently used as tools to monitor landslides



(e.g., Clapuyt et al., 2017; Turner et al., 2015), overland flow erosion (e.g., Eltner et al., 2017; Pineux et al., 2017), river dynamics (e.g., Hemmelder et al., 2018) and vegetation dynamics (e.g., Candiago et al., 2015).

High resolution topography (HRT) data (i.e., point clouds, digital surface models—DSMs or digital elevation models—DEMs) can be acquired using different techniques (e.g., Light Detection and Ranging (LIDAR), Synthetic Aperture Radar (SAR),

Structure from Motion (SfM), etc.). The main advantage of LIDAR and SAR is that both techniques are not limited by illumination or cloud/canopy cover (Passalacqua et al., 2015). However, they are of high operation costs and produce very large datasets that are difficult to interpret. Alternatively, SfM and Multi-view Stereo (MVS) provide low-cost options for acquiring HRT (James and Robson, 2012). The combination of UAV-based aerial photos and SfM algorithm enables to reconstruct three-dimensional (3D) surface models based on a set of digital RGB images from different perspectives (Eltner

et al., 2016). Geomorphic processes can thus be quantified using multi-temporal aerial photogrammetry (Clapuyt et al., 2017; Eltner et al., 2016; Forlani et al., 2018; Turner et al., 2015). However, it is essential to understand the utility, limitations, and particularly the accuracy of HRT derived from these techniques.

Research has shown that the accuracy and precision of SfM-generated HRT data depends on many factors, including camera/image quality, flight plan design, camera modelling methodology, SfM algorithms and georeferencing strategy. SfM

determines the 3D positions of features/points presented in the overlapping part of multiple images by recognizing and matching keypoints, then calculates the exact position and orientation of the camera by performing a fit and minimizing the error through the keypoints (Triggs et al., 2000). Therefore, this process is closely related to the image quality as well as the accuracy and precision of the image georeferencing (James and Robson, 2014b; Mosbrucker et al., 2017).

Camera type and settings are key factors determining image quality and 3D cloud construction (Mosbrucker et al., 2017).

Digital cameras equipped with high quality sensors (e.g., a DSLR camera) provides better image quality due to higher resolution and reduced image noise when compared to a more portable and smaller sensor (e.g., a compact or action camera), which results in high quality DSMs (Eltner and Schneider, 2015; Micheletti et al., 2015). Furthermore, the distance between sensor and surface also determines ground sample distance (GSD), which impacts on accuracy. Eltner et al. (2016) showed in a review of 54 studies that the error of SfM-derived DSMs increased nonlinearly with an increasing surface to camera distance

(Eltner et al., 2016). Another constraint to SfM accuracy is radial distortion and associated calibration of the camera lens (Rosnell and Honkavaara, 2012; Sanz-Ablanedo et al., 2012). The distortion is highly correlated to the lens focal length, which also determines the camera's field of view (FOV). While small focal lengths (or wide-angle) is common to provide a large FOV which could reduce the flight plan density for a given overlap in UAV photogrammetry, these are subject to increased radial distortion and smaller scale for a given object distance, which can degrade accuracy (James and Robson, 2014a;

Mosbrucker et al., 2017). Some studies have investigated the impact of focal length on DEM accuracy (Clapuyt et al., 2016), but mainly on DEM reproducibility. An optimal configuration, corresponding to specific research needs, on the accuracy of aerial photogrammetry remains poorly quantified until now. Furthermore, it should also be noted that there is a large difference



in weight between DSLR (0.5–1.5 kg) and action cameras (0.05–0.15 kg) and this has large implications, not only for flight autonomy (and hence spatial coverage), but also the choice of the UAV platform. Small action cameras can be mounted on small 'micro' drones, which offer the advantage of a much higher portability and applicability, as they are subjected to less stringent UAV flight regulation (e.g., in Belgium, a flexible class 2 UAV operation license allows for a maximum flight height

of 45 m and a weight limit of 5 kg (UAV + payload)).

Camera position and orientation are defined as exterior orientation (EO) parameters. The coordinates of the camera projection center at the exposure time are typically measured by the on-board GNSS receiver, and are then introduced as constraints in the bundle block adjustment, or are used to compute the transformation from the SfM arbitrary reference to the mapping system (Barazzetti et al., 2010). Typically, single GNSS receivers of consumer-grade UAVs are not able to deliver sub-meter level

positioning accuracy. Therefore, the use of ground control points (GCP) surveyed with precise GPS systems or total stations is generally employed for accurate geodetic positioning. The GCP-based georeferencing has been widely proven to be a solid solution for accurate georeferencing (Hawkins, 2016; James et al., 2017; Turner et al., 2016). However, GCPs need to be placed and surveyed and this comes at a cost as it is time consuming. Furthermore, the accuracy strongly depends on the number of GCP's used and their spatial layout (Sanz-Ablanedo et al., 2018). When used in a monitoring study, additional

issues arise from the fact that GCP's can move (weather impact or surface deformations). Finally, a major limitation arises from the fact that GCP's cannot be placed in poorly accessible terrain due to practical or safety reasons (e.g., swamps, landslides or glaciated areas).

Within the last several years, the development of high-quality IMU and GNSS technology and dedicated RTK (Real Time Kinematic) and PPK (Post-Processing Kinematic) solutions for UAVs may provide accurate measurements of the EO

parameters. By double differencing the phase ambiguities between two GNSS/GPS receivers, atmosphere propagation delay and receivers clock errors can be eliminated. RTK positioning requires a stable radio (or internet) link between a base and the UAV, and this can sometimes be challenging due to radio link outages and/or GNSS signal blocks. PPK, in contrast, processes the information after the flight and there is thus no risk of data loss due to link outages. In addition, PPK offers the advantage that positioning can be based on previous and future GNSS data, which can provide a more consistent solution. The utilization

of such an approach has the potential to eliminate the need for GCPs (James et al., 2017). Several studies already investigated the application of RTK/PPK direct georeferencing by the integration of sensor orientation with onboard RTK-GPS (Fazeli et al., 2016; Forlani et al., 2018; Stöcker et al., 2017). In a study performed by Gerke and Przybilla (2016), the block orientation accuracy could significantly be enhanced by using an on-board RTK-GNSS solution. With an enabled RTK-GNSS and cross flight pattern, the best scenario reached a final horizontal geometric accuracy of 4 cm. Recently, both georeferencing methods

are gradually matured and can deliver centimeter-level accuracy in geomorphological applications (Table A1).

The quality of UAV survey output is typically analyzed using the spatial patterns of errors in DSMs, and this includes both the accuracy and the reproducibility of DSM generation. Errors propagate when differences of DSMs (DEM of differences, DoD's)

are computed to quantify topographic change. In order to isolate and quantify the error that is associated with the topographic

reconstructions, reproducibility assessments are critical aspects of monitoring landform changes over time (Brasington et al.,

2000; Wheaton et al., 2010). However, until now the repeatability of direct PPK-based georeferencing for SfM-derived point

clouds and/or DSMs has not been thoroughly evaluated. Past research has shown that a RTK-SfM workflow is repeatable

5   (Forlani et al., 2018), but the flights were done in a very short time frame: i.e. with very similar satellite constellation and light

conditions. It remains uncertain to what extent a PPK-SfM workflow may provide consistent 4D data when survey conditions

are variable, e.g. when monitoring over longer periods of time (e.g. weeks, months). A second issue is the platform: Cheap

RTK-enabled micro-UAV's (small form ca. 25×25 cm, weight 1.4 kg) equipped with small cameras have recently become

available, but their accuracy and repeatability, relative to professional UAV systems (large-form 80×80 cm, weight 4.5 kg)

10  equipped with high-end camera's remains poorly quantified. In particular, the influence of the UAV/camera setup on the

minimum level of topographical change detection should be quantified in order to guide geomorphological applications.

The main objective of this study is to quantify the (i) repeatability, (ii) reproducibility and (iii) efficiency of the PPK-SfM

framework in the context of 4D earth surface monitoring with time-laps structure-from-motion photogrammetry. More

specifically, we aim to (i) assess the accuracy of PPK and Non-PPK solutions in georeferencing to examine the capability of

15  using PPK without the need for GCPs, (ii) assess the reproducibility of surface topography change detection for two different

UAV/camera setups (i.e., with a DSLR camera and an action camera), (iii) investigate the factors controlling the errors

associated with surface reconstructions and (iv) demonstrate the potential of the approach for geomorphological research using

a case-study.

## 2 Material and Methods

### 2.1 Study Site

The study site is located in an agricultural area (1.7 ha) in the Belgium loess belt, ca. 40 km southeast of Brussels, Belgium

(Fig. 1). It is characterized by a slightly undulated terrain with an altitude range between 207 m to 210 m a.s.l. and by very

gentle slopes (mean slope: 1°). The site is partially cultivated while other parts are covered by grass. The surface was classified

into five classes, i.e., bare soil, short grass, shrub, road and hay stacks.

### 2.2 Hardware Setup

#### 2.2.1 Platforms and Payloads

We evaluated (i) a high-payload UAV system equipped with a DSLR camera and (ii) a consumer-grade UAV equipped with

a fisheye action camera. The high-payload aerial system is a custom-built Hexacopter and is equipped with a DJI A2 flight

controller. The platform has an effective payload of 4 kg and an autonomy of ca. 15 minutes. This UAV was equipped with a



Canon EOS 550D camera (18 Megapixels, 5184 × 3456 pixels, with Canon EF 28 mm F/2.8 lens). The consumer-grade UAV

was a DJI Phantom 3 Advanced Drone. We removed the DJI camera/gimbal system and mounted a Hero GoPro 3 camera (12

Megapixels, 4000 × 3000 pixels, with 2.92 mm F/2.8 123° HFOV lens) (Fig. 2). Both platforms are equipped with a compact

multi-GNSS RTK receiver (Reach RTK kit, Emlid Ltd) with RTK/PPK capability as described below.

**2.2.2 PPK-GPS Module**

During the UAV flights, a Reach RS (Emlid Ltd) was mounted on a tripod located in the north of the test area as base station

to provide positioning correction input. The maximal distance between the UAV and the base station was 220 m. The receiver

of the base is configured to log the raw data in a RINEX file at 5 Hz using the satellite GPS, GLONASS and GALILEO. Both

UAVs were equipped with a Reach GNSS receiver to log the raw data as UBX format using GPS and GLONASS satellites.

The antenna model was Tallysman's TW2710, which covers the GPS L1, GLONASS G1, BeiDou B1, Galileo E1, and SBAS

(WAAS, EGNOS, and MSAS) frequency bands. The antenna was mounted on an aluminum plate, with the center right above

the camera lens center to minimize the offset the shift between the antenna phase center and camera projection center. The

antenna height was 22.5 cm and this difference between the antenna and camera projection center was considered during the

post-processing.

For the high-payload UAV, we used the hotshoe of the camera to timemark the pictures with a GPS event that are logged on

a Reach GNSS device mounted on the UAV. As the action camera has no hotshoe, we built an electronic system to integrate

and synchronize the GPS with the action camera. To this end, a single board computer (SBC) is used as a trigger by transmitting

an electrical signal to both the camera and GPS unit. To eliminate the lag between the shutter opening time of the camera and

the GPS recording time, we quantified the delay between the electrical signal and the shutter opening by integrating a LED

light in the circuit. Several delay times were tested until the LED light was visible on the images taken by the action camera.

This procedure resulted in a system where the geotagging was accurately synchronized with the GPS time.

**2.3 Data Collection**

**2.3.1 Flight Planning**

Flight missions were planned using the Autopilot app (Hangar Technology, 2018). The side overlap was set to 80%. The

frontal overlap was defined by the speed of the UAV and the camera trigger interval which was set at 2 s for DSLR camera

and 4 s for action camera, which resulted in a frontal overlap of ca. 90% for both systems.

The flights were performed three times (from late March to early April, 2018) at a constant height of 45 m above the take-off

point leading to a ground sample distance (GSD) of 0.63 cm and 3.11 cm for the DSLR and the action camera, respectively.

The configuration of the three missions was kept constant to test the reproducibility of the resulting DSMs. At the end of the

survey period (6 April), the farmland was plowed leading to a change in surface roughness. To acquire higher GSD and detect

ground surface change, we set the flight height at 35 m for DSLR camera and 20 m for action camera respectively for this mission. Flight mission arrangements are summarized in Table A2. It should be noted that we did not use cross-flight patterns as this may average-out positional errors.

### 2.3.2 Ground Control Points

Sixteen fixed targets were distributed evenly across the study area before the survey as control points (Fig. 2). Depending on the georeferencing methods used (see below), the control points were applied as Ground Control Points (GCP) or Check Points (CP). The targets consisted of a laminated square board (0.3 m × 0.3 m) painted in yellow and a black cross marker in center. They were fixed with nails into the ground and remained at the site for the study period before plowing. For the last flight mission after plowing, new GCPs were deployed and surveyed. The targets were surveyed after each flight mission using a

Reach RS (RTK solution) with the EUREF-IP Network. The correction stream was provided by BRUS station (Brussels, Belgium, Antenna: ASH701945B_M) via NTRIP (Networked Transport of RTCM via Internet Protocol), which had a mean planimetric error of 0.007 m and altimetric error of 0.013 m (https://emlid.com/). The coordinate System was referenced to World Geodetic Datum of 1984 (WGS84).

### 2.4 Data Processing

#### 2.4.1 Georeferencing Configuration

The open source software package RTKLib was used for computing differential positioning (Takasu and Yasuda, 2009). Raw GPS data from the UAV-mounted cameras and the base station were then extracted and corrected by post-processing using RTKLib.

A PPK solution allows various georeferencing configurations by changing the positioning mode in RTKLib, and this enables

us to evaluate the performance of both PPK-GPS or single GPS solutions. To assess the accuracy of different georeferencing options, datasets were processed with four configurations, i.e., *single GPS*, *single GPS + 8 GCPs*, *PPK only*, and *PPK + 1 GCP*. For the conventional methods using GCPs and single GPS, we used the on-board single GPS solution to acquire the images coordinates, and selected half of the targets as 3D GCPs during block control processing. The remaining control points were then used as checkpoints. The setup of GCP/CP is shown in Figure 3. In the *single GPS + 8 GCPs* scenario, the eight

selected GCPs were evenly distributed in the survey area. In the *PPK + 1 GCP* scenario, cross validation was used. We selected one point as a GCP while the remaining targets where then used as CPs and this was repeated sixteen times. The accuracy assessment was based on the average error of the cross-validation.

### 2.4.2 Point Cloud and DSM Generation

The geotagged images were processed with the Pix4D Mapper software (www.pix4d.com). The software uses the SfM algorithm to generate 3D point clouds, DSMs and orthophoto mosaics of the surveyed area. The procedure consists of three main steps: (i) Initial Processing, (ii) Point Cloud generation, and (iii) DSM and orthomosaic generation. First, the photographs are aligned using a point matching algorithm that automatically detects matching points on overlapping photographs and uses these points to simultaneously solve for exterior orientation (EO) parameters. With additional position information that is available for the images or GCPs, the software then georeferences the model and refines the camera calibration by minimizing the error between the modeled locations of the points and the measured locations, meanwhile, non-linear deformations within the model are corrected. Pix4D reports the offset between calculated and measured GCP locations afterwards, providing an initial estimate of model accuracy. After initial processing, Pix4D generates a dense 3D point cloud at a given quality and resolution. In order to highlight the characteristics of the original data, the clouds were not processed or filtered. From these point clouds, DSM and Orthophoto mosaic were then generated. The 3D outputs (i.e., point clouds and DSMs) used for reproducibility assessment were georeferenced using the PPK method (and no GCP were considered). The corresponding GSD of DSM were 0.031 m for action camera and 0.006 m for DSLR camera at the flight height of 45 m.

### 2.5 Data Analysis

#### 2.5.1 Accuracy Assessment

The absolute accuracy test was performed using the CPs (which were not used in the cloud generation) by computing the differences between the coordinates of the checkpoints in the 3D cloud and those measured in the field by GNSS. Mean values and the root mean square error (RMSE) of the differences were computed for each flight to detect systematic shifts and block deformations.

#### 2.5.2 Error Source Exploration

Based on previous work (James et al., 2017), we selected the following attributes to assess factors controlling errors other than GPS error, i.e., tie point density, point cloud roughness, point cloud density and number of images visible. To analyze the factors influencing measurement error, a multiple linear regression analysis was used. Tie point density calculation was based on the tie point cloud which was generated from bundle block adjustment. Point cloud roughness and point cloud density were calculated from the dense point cloud. The number of images was tallied as visible UAV images for a given point position. A dataset of these properties was generated for the points where the error was quantified using the check point positions. Point cloud attributes were calculated using the CloudCompare software while the multiple linear regression analysis was conducted in SPSS.



### 2.5.3 DSM Uncertainty Characterization

To robustly distinguish real changes of DSM/DEM differencing from the inherent noise (Fuller et al., 2003), DoD uncertainty must be considered. Regardless of the approach used to generate DSM/DEMs, the process of accounting for DoD uncertainty follows a consistent progression via three steps: (i) quantifying the error surface ($\delta z$) of each individual DSM surface (ii)

propagating the identified uncertainties into the DoD ($\delta u_{DoD}$) and (iii) assessing the significance of the propagated uncertainty (Wheaton et al., 2010). The tie points differ between each repetition of the survey, and therefore we analyze the error propagation at the DSM level. There are two primary ways to build an error surface. The combined error can be calculated as a single value for the entire DoD based on the average RMSE of each DEM if spatially-explicit estimates of the error do not exist. This method assumes that the errors in each cell are random and independent. Alternatively, a spatially variable error

can be considered for both DEMs independently (e.g., Wheaton et al., 2013). The individual error in the DSMs can be propagated into the DoD as:

$$\delta u_{DoD} = \sqrt{(\delta z_{ref})^2 + (\delta z_{comp})^2} \qquad (1)$$

where $\delta u_{DoD}$ is the propagated error in the DoD, and $\delta z_{ref}$ and $\delta z_{ref}$ are the individual error in referenced DSM and compared DSM, respectively.

Afterwards, the significance of uncertainties in DoD predicted elevation changes are typically expressed. The propagated uncertainties (i.e., $\delta u_{DoD}$) are used to define an elevation change threshold as minimum level of detection threshold (LoD$_{min}$). In practice, a user-defined confidence interval can be applied to indicate the reliability of threshold (e.g., Brasington et al., 2003; Wheaton et al., 2013). If the estimate of $\delta z$ is a reasonable approximation of the individual error, Eq. (1) can be modified to:

$$LOD_{95\%} = \pm 1.96 \left( \sqrt{(\delta z_{ref})^2 + (\delta z_{comp})^2} \right) \qquad (2)$$

Where $LOD_{95\%}$ is the level of detection at 95% confidence interval.

### 2.5.4 Repeatability and Reproducibility Assessment

Using the aforementioned DoD method to assess the repeatability of DSMs, two datasets collected on the same day were selected: (i) DSMs from repeated surveys using the DSLR camera on 5 April; (ii) DSMs from surveys using the action cameras

on 30 March (Table A2). As the earth surface represented by these DSMs was not subject to any change, differences should theoretically be close to zero and have a narrow Gaussian distribution indicating uncertainty/noise inherent to the approach.

### 2.5.5 UAV-based Monitoring of Surface Change

As mentioned above, the bare soil was plowed on 6 April, leading to surface roughness and volume change. Surveys on 5 April and 6 April representing field conditions before and after the plowing were compared to detect the change. A zoomed-

in area as well as a transect was sampled to illustrate the surface change using the LoD$_{min}$ threshold. The sediment budget was



subsequently assessed by computing DoD between DSMs, using the Geomorphic Change Detection (GCD) software (Wheaton et al., 2010).

## 3 Results

### 3.1 Accuracy of The Different Georeferencing Methods

Table 1 summarizes the RMSE ranges in the X, Y and Z directions for the check points for the different block control configurations. For datasets derived from the DSLR camera, the single GPS configuration provided an average planimetric and altimetric accuracy of 2.28 m and 3.93 m, respectively, while the RMSEs for the other three georeferencing configurations were all below 0.03 m. For both the mean error and SD, they were in the same order of magnitude for the *single GPS + GCP*, *PPK* and *PPK + 1 GCP* solutions. For datasets derived from the action camera, the accuracy (mean error) of the single GPS

configuration was 1.53 m in the horizontal and 3.89 m in the vertical. The precision (SD error) of the *single GPS* showed slightly better results for the action camera than the DSLR datasets, i.e., 0.04 m, 0.05 m and 0.12 m in X, Y and Z. When using *single GPS + GCP*, the accuracy was substantially enhanced to cm-level, and the absolute mean errors were less than 0.02 m. The *PPK* and *PPK + 1 GCP* configurations also showed accuracies of ca. 0.03 m. For the *PPK* solution, adding 1 GCP did not affect the accuracy.

### 3.2 Absolute Accuracy of The PPK Configuration

The positional accuracy validation shows a planimetric and altimetric error less than 0.03 m with the DSLR camera and 0.08 m with the action camera when using a PPK solution (Table 2). Both the DSLR and action camera can deliver a mean planimetric error of less than 0.027 m while the altimetric error for the action camera ranges from -0.04 to 0.05 m, which is substantially larger than DSLR surveys (ca. 0.01 m). The maximal RMSE of the vertical error equals 0.076 m for the action

camera, while the RMSE of DSLR surveys are very consistent during the replications and equaled 0.017 m, 0.027 m and 0.022 m.

Analysis of the distribution of check point residuals derived from the surveys, shows that about 40% of the error is within the -0.01 m and 0.01 m interval (i.e., absolute error lower than 1 cm) for the DSLR surveys. A similar proportion was found for the action camera (Fig. A1). The 25th and 75th percentiles were -0.012 m and 0.019 m for the action camera and -0.009 m and

0.015 m for DSLR camera. The check point residuals for the DSLR surveys are distributed more closely around the mean, as all the errors are smaller than 0.05 m. For the data derived from the action camera, the residuals are substantially more dispersed (Table 2, Fig. A1).



### 3.3 Error Correlative Analyses and Spatial Variation

The stepwise multiple regression analyses indicated that the tie point density was significantly correlated to the measurement error for the DSLR dataset ($p$ <0.001), while no correlation with any of the studied variables was found for the action camera dataset. The number of images visible was excluded because of collinearity with tie point density (Table A3). A regression

model was constructed to relate tie point density to error (Fig. 4), and this single factor could explain 42% of the observed variability in error for the DSLR dataset. No other variables were retained in the model. Based on this density-error regression model, spatially explicit error estimates were generated for the repeated surveys with the DSLR camera (Fig. 5). The error is spatially structured with larger errors in shrub areas and smaller errors in non-vegetated areas. The propagated error shows a similar pattern and has a mean value of 0.056 m, which represents the average detection limits of the DSLR camera at this

survey height (45 m) and under the given surface conditions (Fig. 5b).

### 3.4 Repeatability

Figure 6 shows the DoDs when using (i) an average and spatially constant error and (ii) when using a spatialized error map. The histogram shows a unimodal distribution and shows a high repeatability with a mean shift of ca. 0.019 m and a SD of ca. 0.018 m between the two repeated survey outputs. When using a constant error threshold, 94.2% of the area is in the range of

$LoD_{min}$, i.e., -0.042 to 0.042 m (Fig. 6a), and larger differences are visible for the shrub and grassland areas. When using the spatial explicit error for thresholding, a larger error threshold is allocated in vegetated areas due to lower tie point densities, and this results in a relatively tolerant change detection in these regions. When applying a reasonably conservative 95% confidence interval, the DoD is close to zero, indicating a reliable repeatability and reproducibility of the survey.

The DoD map derived from the action camera shows a very different pattern. The 95% confidence interval is ±0.163 m and

has a SD value of 0.039 m, and this is substantially larger than those derived from the DSLR dataset (Fig. 6b). As discussed above, a spatialized error threshold for the action camera could not be derived.

### 3.5 Soil Surface Change Detection

After the repeated surveys, the surface of the study area changed substantially as a result of plowing. The DSMs of the plowed area (before and after plowing) were also analyzed (Fig. 7). It should be noted that the flight height after plowing was reduced

substantially (35 m and 20 m for the DSLR and action camera, respectively). Due to the lower flight heights, the DoD thresholds are similar for both cameras. Although the overall spatial patterns match, the action camera provides a less detailed picture and is not able to detect significant changes in a large part of the study area (Fig. 7a). The zoomed-in area shows a detail of the surface changes along a profile (Fig. 7b). The DSLR camera resulted in much more roughness and level of detail when compared to the profile generated by the action camera. Nevertheless, a significant surface change could be detected for

both approaches when using the $LoD_{min}$ threshold. To illustrate the potential of the approach, we assessed volume changes



over the plowed area of interest using the GCD software (Wheaton et al., 2010) while considering the DoD thresholds. The DSLR dataset evaluated that a total volume of $10.31 \pm 2.70$ m$^3$ was lowered, while $186.06 \pm 76.33$ m$^3$ of the area experienced a volume increase due to changes in bulk density and the construction of ridges. The action camera dataset evaluated the decreased volume at $9.34 \pm 2.36$ m$^3$, while accumulated equaled $203.85 \pm 99.18$ m$^3$ (Fig. 7c). Accordingly, the estimated net

volume of difference by DSLR and action dataset were $175.76 \pm 76.38$ m$^3$ and $194.51 \pm 99.21$ m$^3$, respectively. The similarity in the detected volume changes suggests that the PPK-SfM workflow provides reproducible results.

## 4 Discussion

### 4.1 Accuracy and Precision of PPK Solution in Direct Georeferencing

Our study focused on the accuracy and precision of PPK solutions for direct georeferencing. Results indicated that the DSM

generated by a PPK solution can deliver topographic models with an accuracy below 0.015 m without the use of ground control points (i.e., DSLR camera (EOS): 0.63 cm px$^{-1}$ GSD, 0.011 m planimetric error and 0.012 m altimetric error). The performance of the action camera provides ca. 0.03 m accuracy (Fish-eye lens, 3.11 cm px$^{-1}$ GSD, 0.019 m planimetric error and 0.024 m altimetric error). The PPK approach provided results that are comparable to those that can be obtained using GCPs, i.e., with RMSEs of 0.01–0.02 m. This indicates that direct georeferencing with accurate positioning and orientation is capable to replace

the conventional ground control and acquire centimetric accuracy. As already indicated by previous studies, single on-board GPS are not sufficient for sub-meter level accuracy (Turner et al., 2012a). The quality of GCP-based georeferencing depends on the number and distribution of GCPs (Sanz-Ablanedo et al., 2018). The accuracy is improved by introducing more and densely distributed GCPs, which would induce a tradeoff between survey time and quality of surface reconstruction (Eltner et al., 2016; Smith et al., 2016). Areas with poor distributions of GCPs or lower control precision could be vulnerable to

systematic errors (James et al., 2017). In contrast, precise direct georeferencing of aerial surveys (kinematic GNSS) provides an evenly distributed control framework as each image can be regarded as a control point. In this study, our experiments showed that a high quality GNSS receiver mounted on an aluminum plate that is positioned as far as possible from the UAV electronics can provide reliable accuracy and precision in positioning camera locations. Initial tests showed that the GPS data quality is very vulnerable to interferences from the UAV motors and electronics and special attention should be given to

shielding. The PPK positioning (without a single GCP) of camera positions was shown to provide same level of accuracy as a GCP solution in this study though, there might be biases in the PPK GNSS position estimation due to false solutions which can remain undetected (e.g., false fix in resolving ambiguities). Implementing one GCP did not improve the results in our study, nevertheless, we recommend that using at least one GCP could be the best operational compromise to detect perturbations of the GPS signal. Forlani et al. (2018) balanced the advantage of RTK/PPK and GCP solution and assessed the



*RTK + 1 GCP* configuration, the vertical bias was greatly reduced. However, it should be noted that our system integration did not result in a consistent bias and we did not observe false solutions.

**4.2 Comparison of DSLR and Action Camera**

As for the cameras we used in this study, the main differences were related to the focal length, image resolution and quality.

The action camera with shorter focal length (2.92 mm) provides a larger field of view (diagonal FOV: 115.7°) but is characterized by optical lens distortions. The vertical errors derived from sixteen individual check points were all below 0.07 m indicating that correcting the lens-distortion effects during the image processing greatly eliminated the deformation of the short focal lens (fish-eye) camera. The DSLR camera, due to the larger APS-C sized imagers, higher focal length and higher resolution, together with the complete control of the ISO, shutter speed and aperture settings, produced much less noise and

better overall picture quality. These differences led to better GSD and image contrast. We observed that this assisted greatly in recognizing and matching tie points. For instance, at the same flight height, the DSLR dataset has a higher tie point density (mean: 213.5 points m$^{-2}$) than action camera (mean: 12.1 points m$^{-2}$) and the detailed images help finding and matching tie point. However, the large FOV of the action camera has also some advantages that may to some extent compensate the image quality difference. Using a fisheye camera allows larger angles between homologous rays and convergent imaging that may

minimize systematic errors (e.g., Wackrow and Chandler, 2011; James et al., 2017) or may provide a better determination of the internal geometry camera calibration, if appropriate parameters are introduced in the bundle adjustment. This feature may explain the observed absence of a significant correlation between error and tie point density for the action camera (Table A3). We performed an additional analysis to verify this by only using half of the images (i.e., we removed one out of every two images) of the action camera dataset: the results showed that the error (based on sixteen control points) did not increase when

using a substantially decreased tie point density.

Our result showed that a correlation exists between tie point density and accuracy for the low FOV DSLR dataset. Although the number of images was excluded in the multiple linear analyses, we found this factor to be strongly correlated to tie point density (Fig. 8). In contrast to the low focal length action camera with a large FOV, a single tie point is normally captured by only 7-12 images, where more tie points are found in the intersecting area while less are found below the image positions. This

results in a structured pattern in tie point density and suggests that an increase in survey (images) density may further improve the accuracy when using RTK/PPK solutions. However, a substantial part of the observed error still remained unexplained. Since we could not identify other controlling factors, we postulate that these errors are random errors, which are constrained by the GSD, color and contrast, and limitations of the SfM algorithm.

To visualize the two camera setup outputs and assess the potential of soil roughness measurement in different surface type, we

derived two representative transect (Fig. 7b). Due to a higher GSD, the DSLR-derived data showed abundant and sharp details, while data from the action camera were relatively smooth. It should be noted that due to the large FOV, the action camera





required a flight plan that was much less dense than for the DSLR camera (about half), indicating that a much larger area (about double) could be surveyed in the same time. However, this larger spatial coverage comes at the cost of ground resolution. A lower distance between the camera sensor and the surface is required for the action camera to obtain the same GSD as DSLR camera (for the GoPro and EOS cameras used in this study, the flight height ratio to obtain the same GSD equals 1:3.5). Our

surveys at different flight heights indicated that a better accuracy and precision can be obtained with lower flight height (Table 2), which was also reported by Eltner et al. (2016). However, lower flight height means a denser flight plan and lower flight speed for a given constant side/front overlap. We calculated the required survey time for both cameras to obtain the same GSD, image overlap and survey area by adjusting flight height and speed. To obtain the same GSD, the action camera requires a lower flight altitude while at the same time, its larger FOV allows to space the flight lines more widely, relative to the DSLR

camera. For a same GSD, the time ratio of the DSLR to action camera was ca. 1:1.5 for the camera models used here. In practical UAV survey, it is crucial to take the sensor weight and size into account, as well as the payload and endurance of UAV platform. Proper selection of devices depends on specific application scenarios, UAV regulations and accuracy/precision requirements. Taking advantage of the large FOV of the compact action camera, it is feasible to cover more area at the cost of GSD and accuracy.

**4.3 Application of UAV-SfM Framework for Surface Change Detection**

This study showed that both UAV-camera setups used here resulted in similar data to quantify 3D topography changes. In this study, the $LoD_{95\%}$ for the DSLR camera was ca. ±0.08 m and while it was ca. ±0.16 m for the action camera at a flight height of 45 m. Lower flight altitude, and hence a higher GSD resulted in a better change detection. For a flight altitude of 20 m, the $LoD_{95\%}$ for the action camera was reduced to ca. ±0.09 m. Areas where the tie point density was low resulted in a poor DSM

reconstruction. Therefore, the generation of DSMs depends strongly on the characteristics of the measured surface. It is important to understand the effects of different types of surfaces on the SfM output, particularly in a region with a complex surface, including vegetation and rough objects. Vegetation has long been recognized as a source of error in photogrammetry (Lane et al., 2000; Messinger et al., 2016), and in this study the vegetated areas show larger errors than bare soil (Fig. 5). For dense vegetation with cluster of leaves, the wind caused movement and illumination change and this increased the complexity

of the imagery, leading to difficulties in isolating tie points (Harwin and Lucieer, 2012). We showed that carrying out topographic change detection using spatially explicit error thresholds improves the reliability and sensitivity because it takes into account the factors controlling the error.

Our study showed that the PPK positioning is a robust solution for monitoring surface change and estimating sediment budgets at very high spatial and temporal resolution. This technique can be very advantageous when it comes to monitoring large areas

that are poorly accessible or require repeated surveying (Clapuyt et al., 2017; Eltner et al., 2016).

A relatively cheap RTK/PPK-enabled micro UAV (small-form 25×25 cm, weight 1.4 kg, autonomy 15 minutes) provided similar accuracy and repeatability as a professional multirotor UAV system (large-form 80×80 cm, weight 4.5 kg, autonomy 15 minutes). Based on our analysis, we suggest that using a micro-drone/action camera setup is suitable for large scale monitoring (e.g., gully erosion, landslides, glaciers, etc.) when high GSD are not needed. Furthermore, in countries with strict

UAV regulations and/or inaccessible regions (e.g., mountains) a light-weight system can be more easily transported on the field than a large UAV system. The DSLR camera setup can be used when high resolution is needed, for example for soil roughness assessment, sheet and tillage erosion, solifluction, riverbank erosion, etc. Finally, a key step in PPK positioning is to obtain GPS data from a statoinary base station. In this study, we used an internet-enabled system to geolocate the base station for each flight. For areas where an internet is absent or unreliable, and long-term monitoring is required, we suggest to

set up a permanent reference point that can be used to position a local base station (e.g., a concrete pole).

**5 Conclusion**

The UAV-SfM framework is increasingly used in geomorphology to accurately capture the Earth's surface. Our study showed that the application of PPK (Post-Processing Kinematic) in direct georeferencing can provide cm-level accuracy and precision with greatly improved field survey efficiency, and this without the need to survey a single GCP. We investigated the positional

accuracy and the repeatability of DSMs by repeating the same flight plans. The PPK solution had a similar accuracy (mean: ca. 0.01 m, RMSE: ca. 0.03 m) as the traditional approach using georeferencing based on GCP's. We also showed that camera properties (i.e., focal length, resolution, sensor quality) have a large impact on the accuracy. The DSM reconstruction and surface change detection based on a DSLR and action camera were reproducible: the main difference lies in the level of detail of the surface representations. We found that tie point density was significantly related to the error of the topography

reconstruction for the DSLR camera. By exploiting the relation between error and tie point density, we demonstrated that a spatially explicit DoD threshold, can greatly improve surface change detection. Overall, the PPK-SfM workflow overcomes the limitations of GCP, providing a high-precision and high-efficiency solution in surveying and geomorphological applications.

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





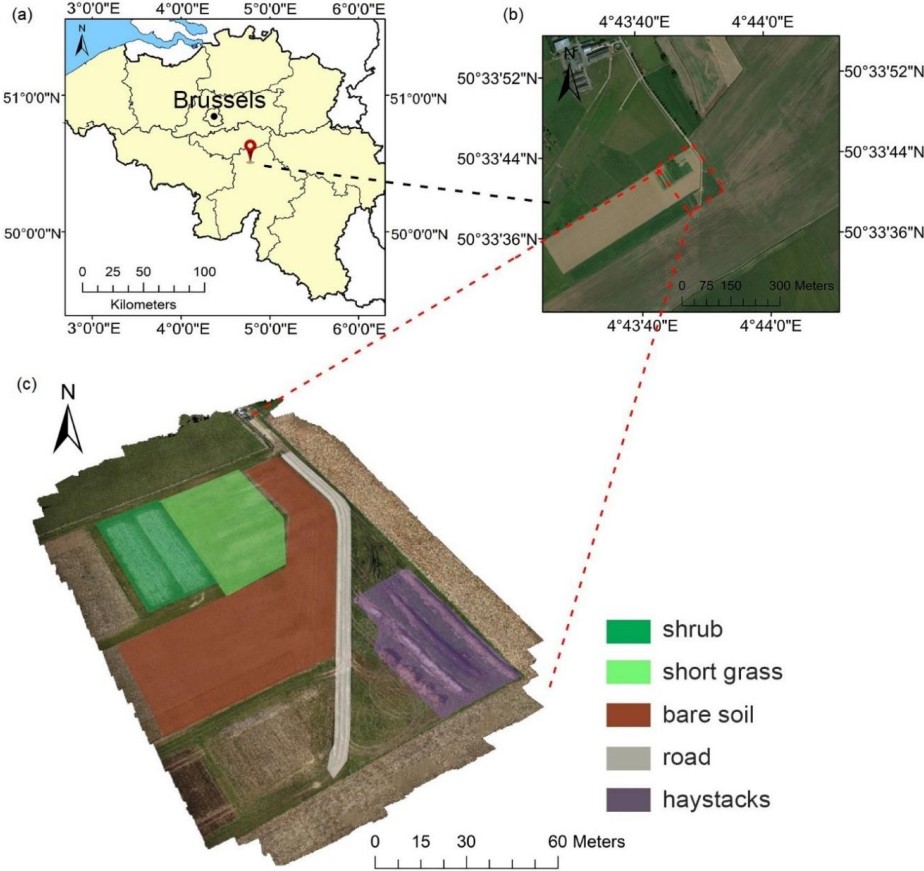

Figure 1: Description of the study sites. (a) location of study site (b) satellite image of the study site (c) classification of the surface used in the analysis.



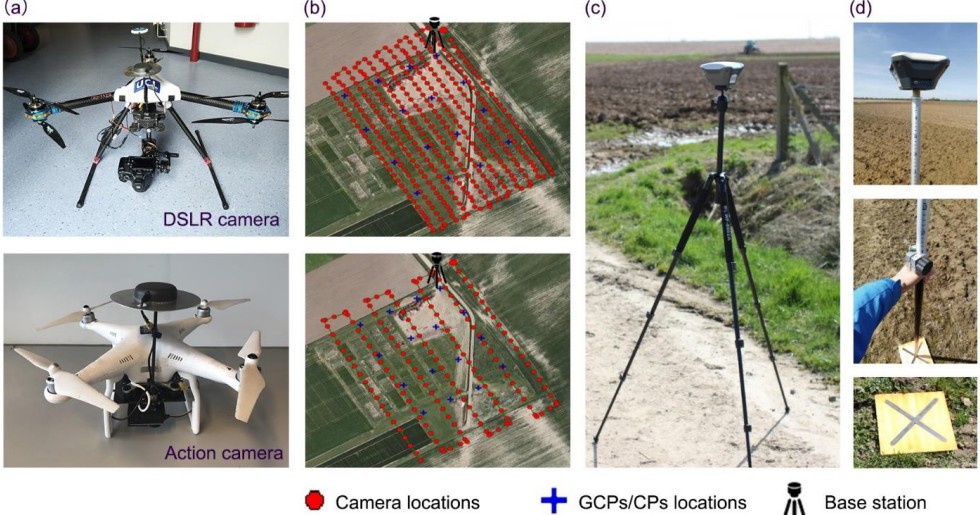

**Figure 2: Experimental setup: (a) UAV/camera setup: DSLR camera (EOS 550D) mounted on RPAS Type Y6, Action camera (GoPro Hero 3) mounted on a Phantom 3 Advanced (b) Flight plan (Top: RPAS Type Y6 with DSLR camera; Bottom: Phantom 3 with action camera) and images and GCPs/CPs distribution (c) Field base station and (d) Measurement of GCPs/CPs.**




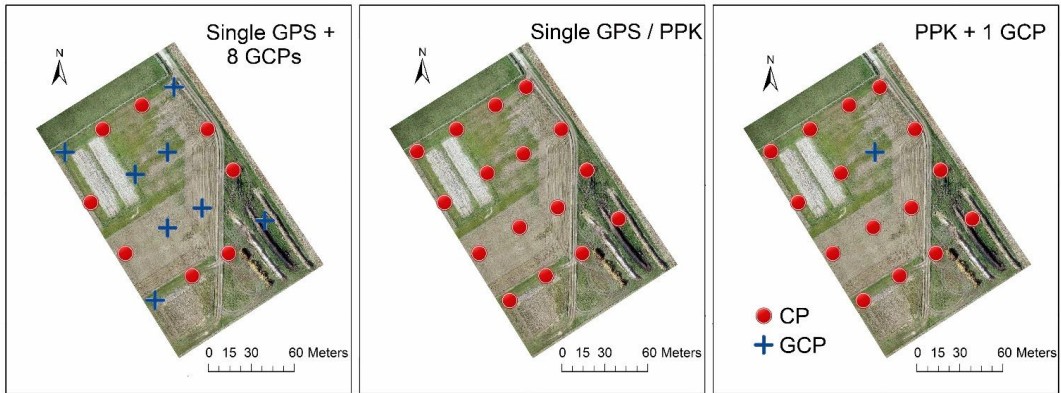

**Figure 3: Distribution of GCPs and CPs and illustration of the different georeferencing configurations:** *Single GPS*, *Single GPS + 8 GCPs*, *PPK*, *PPK + 1 GCP*. **Note: Cross validation was implemented in** *PPK + 1 GCP* **configuration, i.e., one single control point was used as GCP in each processing.**





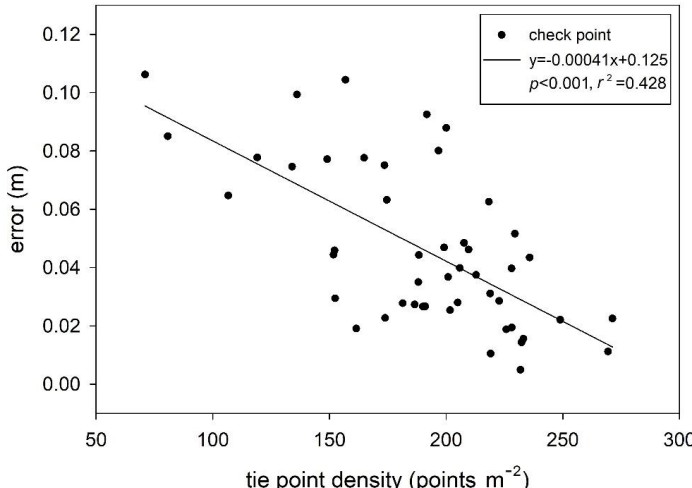

**Figure 4: Linear regression between the check point residuals (XYZ error) and the corresponding local tie point density (Dataset: DSLR camera surveys of flight height at 45 m)**



(a)

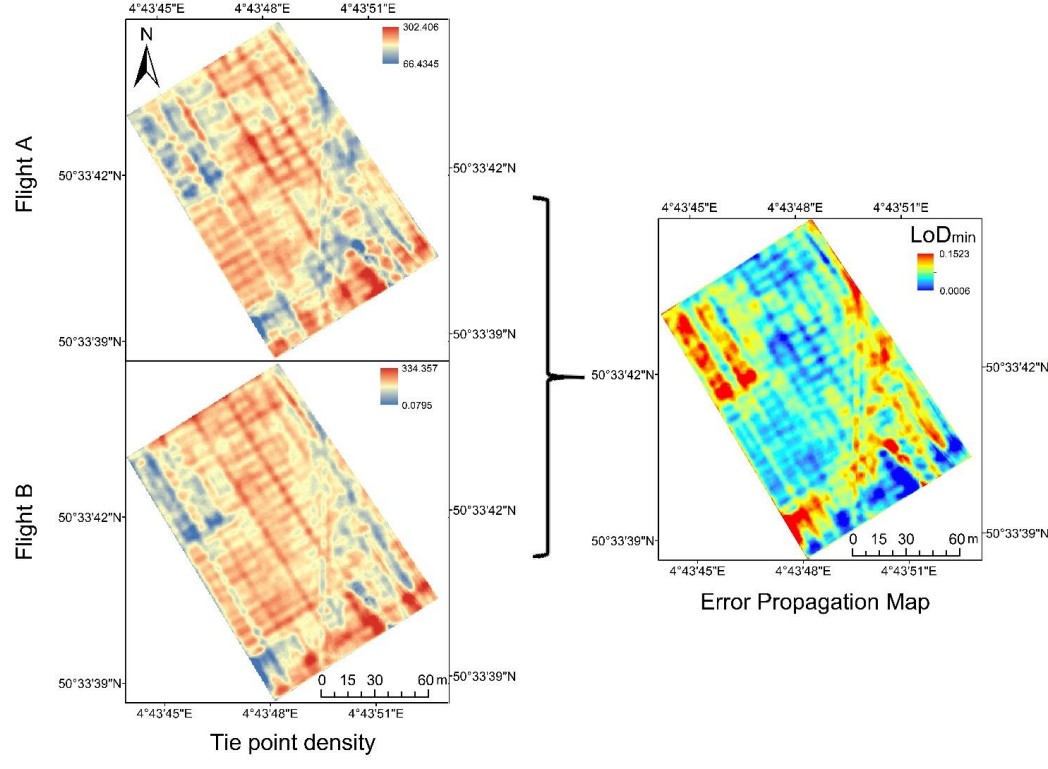



(b)

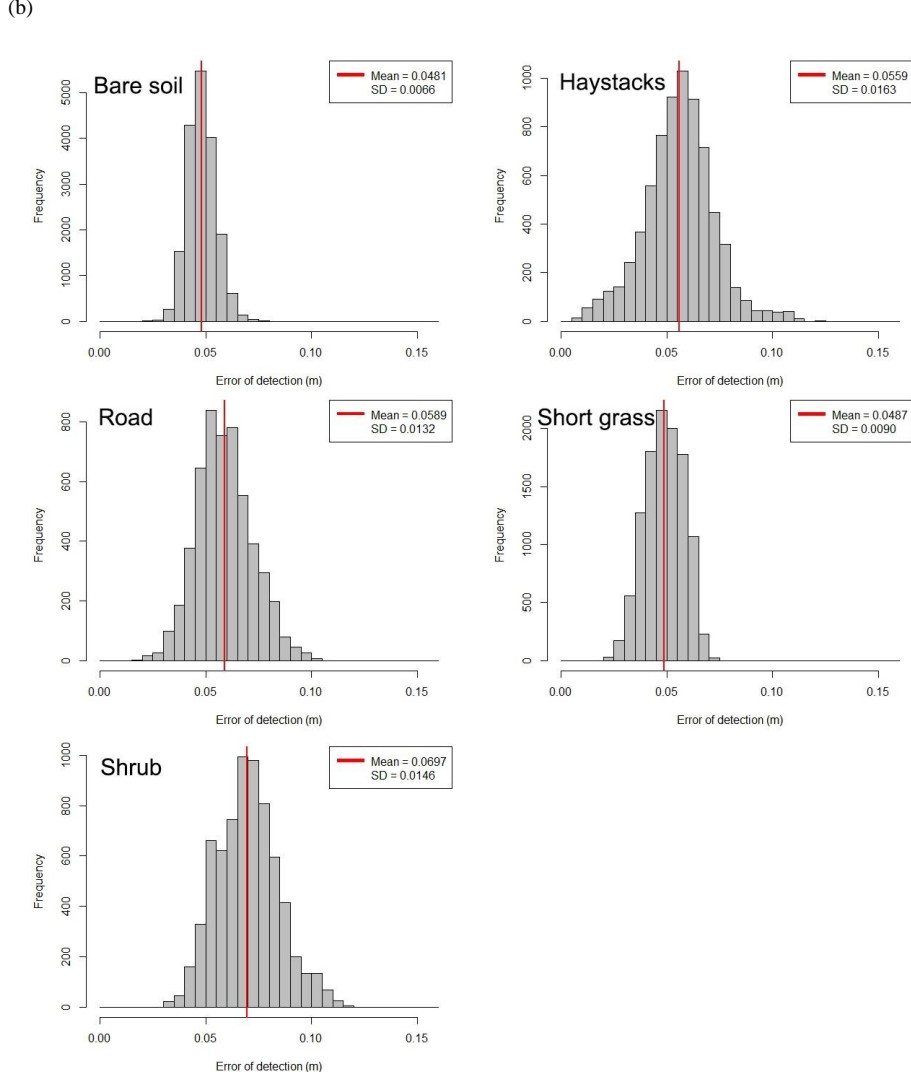

**Figure 5: Illustration of tie point density, the propagated error maps of the repeated surveys based on the DSLR camera on 5 April.**

**(a) Tie point density and propagated error map. The error was estimated by the density-error regression model (y = -0.00041x +**

5  **0.125). (b) Histogram of propagated error for different surfaces.**





(a)





(b)

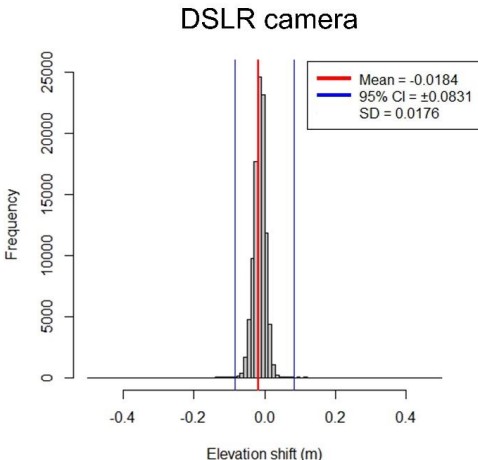
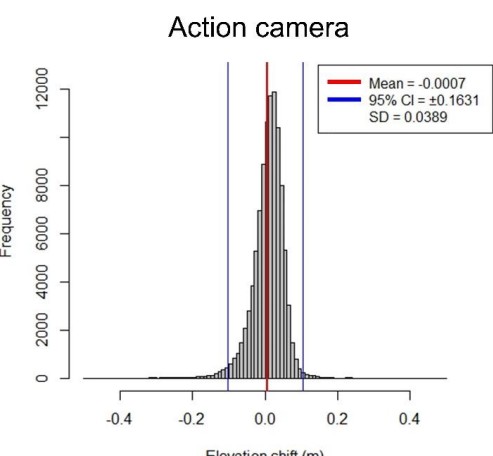

**Figure 6: Illustration of the calculation of DoD for DSMs between two repeated surveys by DSLR camera on 5 April and by action camera on 30 March, respectively. (a) Original DoD and its classification by constant and spatialized error. The constant error was calculated from check point residuals (RMSE) of each DSM. The spatialized error was based on tie point density regression model. (b) Histogram shows distribution of elevation shift values of DSM subtraction. Thresholds were at a 95% confidence interval (LoD$_{95\%}$) and LoD$_{min}$, respectively.**



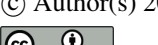

(a)

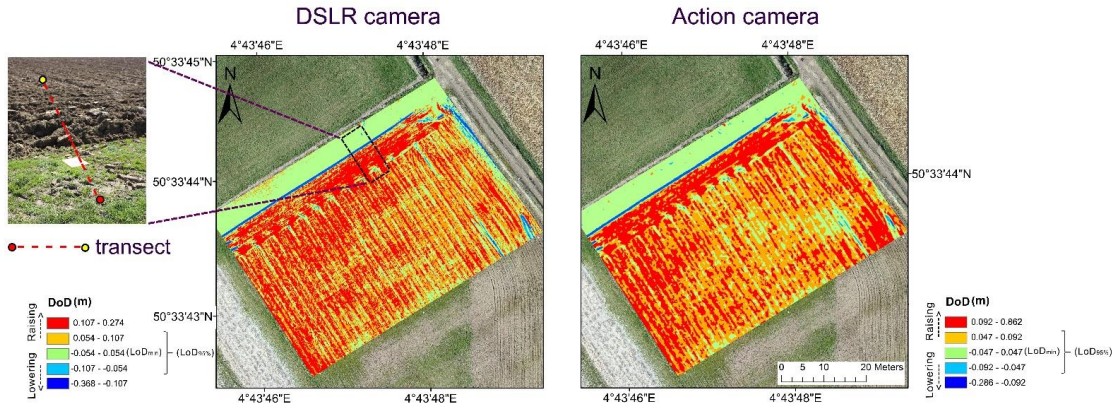

(b)

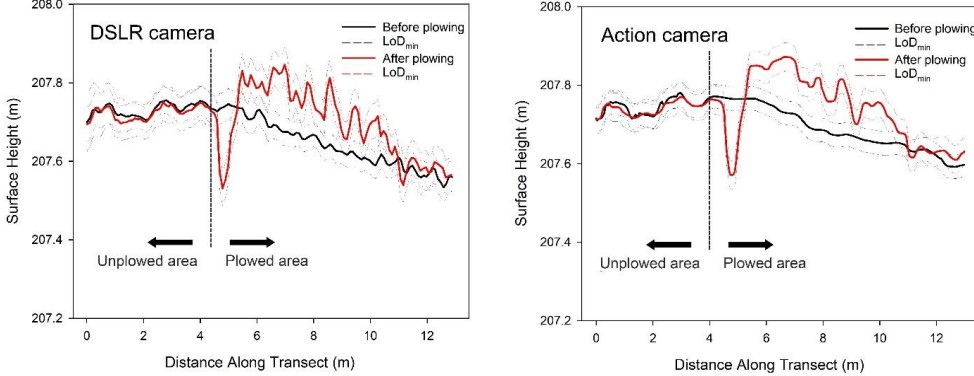

(c)

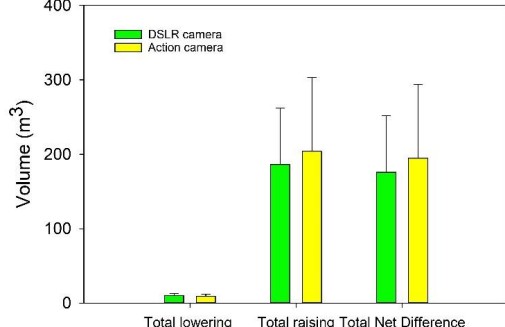

**Figure 7: Change detection based on DoD (datasets: DSLR and action camera surveys on 5 April and 6 April). (a) Surface change map (b) Height profiles sampled at identical location from the corresponding DSMs before and after plowing. Line graph shows height profiles along the sample transect (X-axis: position along the transect, Y-axis: surface height) (c) Volumetric sediment budget.**


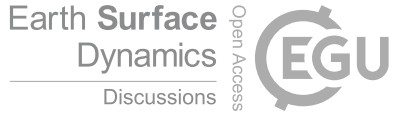

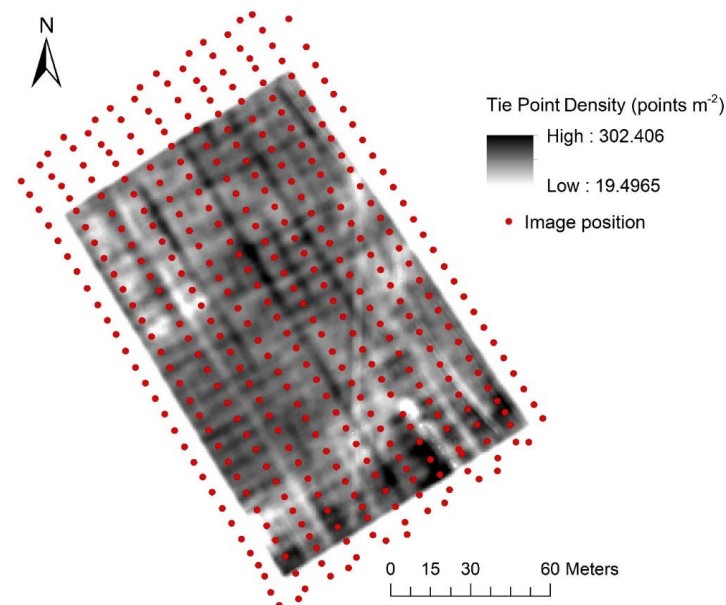

**Figure 8: Tie Point density and image position. Dataset: DSLR camera survey on 5 April.**



**Table 1. Mean errors (ME), Standard deviation (SD) and Root mean square errors (RMSEs) on check points respectively for horizontal and vertical coordinates, for the different configurations for each of the three flights (flight height at 45 m)**

| Camera | Georeferencing method | Mean(m) | | | | SD(m) | | | | RMSE(m) | | | |
|---|---|---|---|---|---|---|---|---|---|---|---|---|---|
| | | X | Y | XY | Z | X | Y | XY | Z | X | Y | XY | Z |
| DSLR camera (EOS) | Single (0 GCP) | -0.538 | -2.212 | 2.276 | 3.927 | 0.144 | 0.144 | 0.204 | 0.180 | 0.557 | 2.216 | 2.285 | 3.931 |
| | Single + GCP (8 GCP) | 0.003 | -0.001 | 0.003 | 0.011 | 0.013 | 0.019 | 0.023 | 0.023 | 0.013 | 0.019 | 0.023 | 0.026 |
| | PPK (0 GCP) | 0.005 | 0.014 | 0.015 | 0.003 | 0.013 | 0.014 | 0.019 | 0.022 | 0.014 | 0.020 | 0.024 | 0.022 |
| | PPK + 1 GCP | 0.007 | 0.011 | 0.013 | -0.019 | 0.014 | 0.014 | 0.020 | 0.022 | 0.015 | 0.018 | 0.023 | 0.029 |
| Action camera (GoPro) | Single (0 GCP) | -0.104 | -1.524 | 1.528 | 3.884 | 0.041 | 0.050 | 0.065 | 0.124 | 0.112 | 1.525 | 1.529 | 3.886 |
| | Single + GCP (8 GCP) | -0.001 | 0.002 | 0.002 | -0.018 | 0.006 | 0.011 | 0.012 | 0.014 | 0.006 | 0.011 | 0.013 | 0.023 |
| | PPK (0 GCP) | 0.004 | 0.010 | 0.011 | 0.012 | 0.016 | 0.016 | 0.023 | 0.017 | 0.018 | 0.019 | 0.026 | 0.042 |
| | PPK + 1 GCP | 0.007 | 0.007 | 0.010 | -0.026 | 0.013 | 0.012 | 0.018 | 0.017 | 0.015 | 0.014 | 0.020 | 0.032 |



**Table 2. Assessment of positional accuracy based on check point (n = 16) residuals for the eight sets of flights with PPK mode.**

| Camera | Flight Mission Date | Flight Height (m) | Mean(m) | | | | RMSE(m) | | | |
|---|---|---|---|---|---|---|---|---|---|---|
| | | | X | Y | XY | Z | X | Y | XY | Z |
| DSLR camera (EOS) | 21, March | 45 | 0.011 | -0.011 | 0.016 | 0.012 | 0.016 | 0.015 | 0.022 | 0.017 |
| | 30, March | 45 | 0.009 | 0.013 | 0.015 | 0.016 | 0.028 | 0.021 | 0.035 | 0.027 |
| | 05, April | 45 | 0.005 | 0.014 | 0.014 | 0.003 | 0.014 | 0.020 | 0.024 | 0.022 |
| | 06, April | 35 | 0.006 | 0.000 | 0.006 | 0.019 | 0.008 | 0.004 | 0.008 | 0.023 |
| Action camera (GoPro) | 21, March | 45 | 0.018 | 0.021 | 0.027 | 0.049 | 0.028 | 0.031 | 0.042 | 0.076 |
| | 30, March | 45 | 0.008 | 0.008 | 0.011 | -0.042 | 0.016 | 0.015 | 0.022 | 0.051 |
| | 05, April | 45 | -0.010 | 0.008 | 0.013 | 0.009 | 0.014 | 0.013 | 0.019 | 0.024 |
| | 06, April | 20 | -0.013 | 0.020 | 0.024 | -0.015 | 0.018 | 0.024 | 0.030 | 0.020 |





**Appendix A**

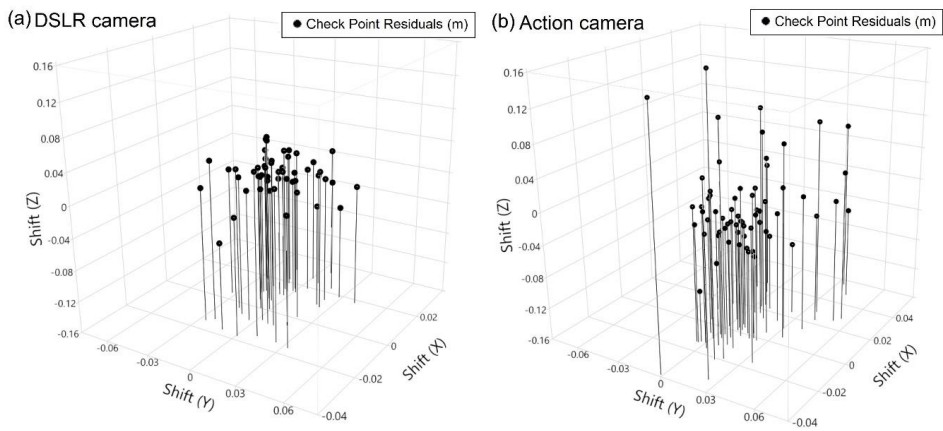

**Figure A1. Distribution of positional error based on check points assessment. 3D scatter plots show shifts on X, Y, and Z of each check point. (a) Dataset: DSLR surveys of flight height at 45 m (b) Dataset: action camera surveys of flight height at 45 m.**





**Appendix B**

**Table A1. Summary of positional accuracy assessments conducted in various published studies.**

| Authors | Validation Method | Flight height and GSD | Motivation/Application | Accuracy | Georeferencing Method |
|---|---|---|---|---|---|
| (Turner et al., 2012b) | Check points | 50 m, ca. 1 cm px⁻¹ | Accuracy assessment | Mean absolute horizontal accuracy of 0.66 and 1.25 m | Direct georeferencing with single GPS |
| (Harwin and Lucieer, 2012) | Check points | 50 m, ca. 1 cm px⁻¹ | Coastal erosion | Mean absolute horizontal accuracy of 0.10 and 0.13 m | GCP |
| | Check points | 30–50 m, 1–3 cm px⁻¹ (after down-sampling) | | Horizontal RMSE of 0.001–0.083 m / Vertical RMSE of 0.04–0.06 m | GCP |
| (Ouédraogo et al., 2014) | DEM of difference | Maximum of 100 m, 3.3 cm px⁻¹ | Agricultural soil microtopography | Mean absolute difference of 0.074 m | GCP |
| (Uysal et al., 2015) | Check points | 60 m, 5.2 cm px⁻¹ | Accuracy assessment | Mean vertical accuracy of 0.062 m (from altitude of 60 m) | GCP |
| (Fazeli et al., 2016) | Check points | 120 m, 2.38 cm px⁻¹ | Accuracy assessment | Mean horizontal accuracy of 0.132 m / Mean vertical accuracy of 0.203 m | Direct georeferencing with RTK-GPS |
| (Clapuyt et al., 2016) | DEM of difference | 50 m, 0.43–0.77 cm px⁻¹ | Reproducibility assessment | Mean absolute error of 0.06 m | GCP |
| (Stöcker et al., 2017) | Check points | 100 m, 2.8 cm px⁻¹ | Accuracy assessment | Mean accuracy on X, Y and Z: 0.217, 0.186 and 0.053 m | Direct georeferencing with RTK-GPS |
| (Glendell et al., 2017) | DEM of difference | 23–40 m, 0.6–1.1 cm px⁻¹ | Upland soil erosion | RMSE of DoD from 0.05 m to 0.35 m | GCP |
| (Forlani et al., 2018) | Check points | 90 m, 2.3 cm px⁻¹ | Accuracy assessment | Mean horizontal accuracy of 0.024 m / Mean vertical accuracy of 0.046 m | Direct georeferencing with RTK-GPS |
| | Check points | 90 m, 2.3 cm px⁻¹ | | Mean horizontal accuracy of 0.015 m / Mean vertical accuracy of 0.023 m | GCP |
| | DEM of difference | 90 m, 2.3 cm px⁻¹ | | Mean absolute difference of 0.125 m | Direct georeferencing with RTK-GPS |
| (Eker et al., 2018) | Check points | 40 m, 0.72–0.89 cm px⁻¹ | Monitoring landslide | RMSE of 0.04 m | GCP |
| (Rossini et al., 2018) | Check points | 110 m, 4.3–4.5 cm px⁻¹ | Tracking glacial dynamics | Total RMSE of 0.153 m | GCP |
| (Duró et al., 2018) | Check points | 25 m, 2.1 cm px⁻¹ | Bank erosion | Mean error of -0.05–0.04 m | GCP |





**Table A2. Overview and key parameters of flight missions**

| | Date | Camera | Flight Height (m) | Speed (m s$^{-1}$) | Area Covered (ha) | Satellite PDOP value | Ground Sampling Distance (cm px$^{-1}$) | Number of Images |
|---|---|---|---|---|---|---|---|---|
| **Before plowing** | 21.03.2018 | EOS | 45 | 3.4 | 3.60 | 1.5 | 0.63 | 346 |
| | | GoPro | 45 | 3.4 | 2.05 | 1.2 | 3.11 | 154 |
| | 30.03.2018 | EOS | 45 | 3.4 | 1.69 | 1.3 | 0.63 | 323 |
| | | GoPro* | 45 | 3.4 | 2.05 | 1.4 | 3.11 | 140 |
| | 05.04.2018 | EOS* | 45 | 3.4 | 1.69 | 1.2 | 0.63 | 314 |
| | | GoPro | 45 | 3.4 | 2.05 | 1.5 | 3.11 | 137 |
| **After plowing** | 06.04.2018 | EOS | 35 | 3.0 | 0.85 | 1.3 | 0.48 | 225 |
| | | GoPro | 20 | 2.6 | 1.01 | 1.2 | 1.25 | 108 |

* Flight mission was executed twice, marked as repeat_A and repeat_B





**Table A3. Results of multiple linear regression analysis. (a) Dataset: DSLR camera surveys of flight height at 45 m, regression method: "stepwise" (b) Dataset: Action camera surveys of flight height at 45 m, regression method: "enter". Note: "stepwise" method includes or removes one independent variable at each step, based (by default) on the probability of F (p-value). "enter" method forces variables introduced in one step in order of decreasing tolerance. For action camera dataset, no variables were entered the regression equation using "stepwise" method. Therefore, the variables were introduced by "enter" method.**

(a)

Model Summary:

| Variables Entered | $r$ | $r^2$ | Std. error of the estimate |
|---|---|---|---|
| Tie point density | 0.654 | 0.428 | <0.001 |

Coefficients:

| Model | Standardized Coefficients | t | Sig. |
|---|---|---|---|
| (Constant) | - | 9.094 | <0.001*** |
| Tie point density | -0.654 | -5.868 | <0.001*** |

Excluded Variables:

| Model | t | Sig. | Partial Correlation | Collinearity Tolerance | VIF |
|---|---|---|---|---|---|
| Number of images | -0.934 | 0.354 | -0.122 | 0.457 | 2.19 |
| Dense cloud roughness | -1.262 | 0.212 | -0.163 | 0.983 | 1.017 |
| Dense cloud density | 1.275 | 0.208 | 0.165 | 0.978 | 1.022 |

***Significant at 0.01

(b)

Model Summary:

| $r$ | $r^2$ | Std. error of the estimate |
|---|---|---|
| 0.276 | 0.076 | 0.0179 |

Coefficients:

| Model | Standardized Coefficients | t | Sig. |
|---|---|---|---|
| (Constant) | - | 1.887 | 0.069 |
| Tie point density | -0.107 | -0.495 | 0.624 |
| Number of images | -0.152 | -0.616 | 0.542 |
| Dense cloud roughness | -0.260 | -1.429 | 0.164 |
| Dense cloud density | 0.110 | 0.520 | 0.607 |