# Peer review of "Evaluating the potential of Post-Processing Kinematic (PPK) georeferencing for UAV-based Structure from Motion (SfM) photogrammetry and surface change detection"

_Earth Surface Dynamics, 2019_

## Referee Comment (RC1) · Mike James (Referee) · 18 Feb 2019

General comments: The paper presents analyses of topographic data acquired by UAV and SfM-MVS photogrammetry using a PPK-GNSS direct georeferencing approach. This is a technique with broad relevance to a wide range of disciplines because the method will become increasingly widespread. Nevertheless, novel findings within the work are difficult to identify clearly and I haven't found the methods section sufficiently detailed to fully understand what has been done. The contribution of the work would be much clearer if existing similar work was evaluated more critically to provide a detailed context, and the aims and outcomes more concisely defined. Drawing more

deeply on published work should allow statements of well-established principles (such as "camera properties ... have an impact on the accuracy") to be removed from the key sections such as abstract, discussion and conclusions so that the new findings can be more clearly communicated. The work is of interest but insufficiently described and, currently, the paper is somewhat challenging to assimilate. Overall, my suggestions below are aimed at highlighting the most transferrable new results from the work, by downplaying areas that have been previously covered and extending discussion to explore the underlying concepts further.

Specific comments: With the paper focussed on PPK direct georeferencing for UAV surveys, the introduction would be well served by focussing on this. With UAV-SfM approaches not being so new, substantial regions of text (e.g. up to P 3), which introduce the broader aspects and uses of UAV photogrammetry could be condensed into a few sentences or a single paragraph. The introduction would be strengthened by incorporating Table A1 into the main text and critically evaluating the progress of PPK-controlled UAV surveys so far. Consideration of established use of this approach for crewed aircraft could be covered briefly. Inclusion of the recent PPP work by Grayson et al. (2018; DOI: 10.1111/phor.12259) – and references included within it – will also strengthen this section.

One aspect of the work is an exploration of predictors for survey repeatability. The rationale for some of the selections could be strengthened here, and the utility of tie point density (which is shown to explain <50% of the variance for one camera) more critically considered. How useful is this, given that the analysis only appears to work for one camera and requires deployment of GCPs to determine the relationship? The number of tie points retained per image is usually a software setting that can be varied. Consequently, any parameterisation of repeatability would be software and UAV system dependent. Furthermore, other more important parameters are not considered. Within the bundle adjustment, measurement precision for a tie point is related to the number and angles of observations – how do these vary? The authors cite James et

al. (2017) who show how point coordinate precision varies spatially and can be linked directly to these photogrammetric factors and other georeferencing factors within the adjustment. Consequently, maps of 3D point precision can be determined without any GCPs. The work here would be strengthened by discussing the authors 'spatialised error' approach in context with the 3D precision maps of James et al. (2017). The authors could also consider the findings of Mosbrucker et al (2017; DOI: 10.1002/esp.4066) within the discussion (or introduction).

Comparison of results from different cameras (S4.2; particularly the last paragraph) dominantly reflects established relationships between camera/flight parameters and conventional survey design principles. In my view, this material should form the rationale behind the survey design, and be given within the introduction or methods sections. Placing this within the discussion detracts from the newer aspects of the work (the PPK). Throughout, when discussing results from different cameras, I suggest that a dimensionless approach based on pixels (or ground sampling distance) is also used. This could be used to assess the quality of the photogrammetric networks achieved, and to generate insight – again, see previous work, including that of Mosbrucker et al. (2017). I would actually see a much more detailed assessment of the PPK performance as providing the most useful (i.e. transferrable) insight.

I have annotated the ms with areas where I have been unclear about the methods. Unfortunately, this means I may not have fully understood all the aspects of the results. It would be good to see some more details to support the photogrammetric processing though – e.g. what were the rms image residual magnitudes? Did they vary image-to-image in any way that would help understanding of the repeatability? The clear image overlap outlines shown in Fig 6a suggest that camera positions may have been over-constrained in at least one survey (e.g. see a similar effect in Fig 1 of James et al. 2017a - http://dx.doi.org/10.1016/j.geomorph.2016.11.021, resulting from over-weighting the GCPs in that case). Details of the a priori assigned camera position precisions used in the adjustments need to be provided and, given that they are often

optimistically estimated, the effects of diluting the estimates could be explored.

Technical corrections, typing errors, etc.: Most of the detailed suggestions have been annotated on the manuscript, with a few additional points below. The text does contain typos and errors in English that I have not had the opportunity to individually correct.

Fig 2 – Are photographs of the GPS system etc. really needed (c, d)? Much more valuable would be examples of the imagery processed (i.e. the underpinning data on which the work relies), with enlarged excerpts to illustrate the image quality and show how the GCPs have been imaged.

All figures need to be checked for readability of the text labels. In particular, all map figures have scale and other labels which are far too small to be readable, and font sizes vary substantially across the figures. Labels must be readable: more consistent font sizes, of at least 9 point, will help.

Fig 4 Rephrase 'error of detection' for clarity.

DoDs – represent image overlaps etc for the DSLR but not for the action camera.

Fig 7b LoD before/after lines indistinguishable – needs more careful visualisation.

Fig. 8 Colour scale given to four decimal places could be tidied up.

Table 1 The caption mentions three flights but I can only see data from two (i.e. one with DLSR, one with action camera). Which flights are these? Where are the results from the others?

Table 2 The link to Table 1 is unclear. 05 April DLSR results are the same as in Table 1, but no similar repetition for Action camera. Maybe I haven't understood what Table 1 is?

Fig A1. I am not convinced how useful these visualisations are – it is difficult to extract much from them. I would suggest that a more informative plot would be as an XY map of discrepancy vectors, with symbols to indicate the check point position and Z-

discrepancy showed by symbol colour. This way, any spatial systematics (which would be concealed in the current plots) would be clear.

Table A2 This information is critical to a reader's understanding; it needs to be early in the main manuscript, not in the appendices. Why were some flights repeated? Where are these repeated data, and what did they show?

Please also note the supplement to this comment:
https://www.earth-surf-dynam-discuss.net/esurf-2019-2/esurf-2019-2-RC1-supplement.pdf

**Supplement:**

[revised manuscript text omitted]

---

## Referee Comment (RC2) · Anonymous Referee #2 · 28 Feb 2019

Dear Editor,

The paper "Evaluating the Potential of PPK direct Georeferencing for UAV-SfM Photogrammetry and Precise Topographic Mapping" fits the scope of the journal and I consider that the paper is very interesting for the Earth Surface Dynamics' readership. Moreover, it is a well-written paper, with very interesting results and rigorous validations. However, some minor revisions and comments must be fixed before the final publication:

Comment 1: Introduction (section 1) and Discussion (section 4.2): There is a very recent publication where it is compared the accuracy of different PPK approaches and

other positioning alternatives, using DLSR cameras (10.1016/j.jag.2018.10.018). This could be in the introduction and in the discussion, since this research follows a similar workflow.

Comment 2: P6 (section 2.3.2): Why did you not post-processed the static GNSS measurements?

Comment 3: P7 (section 2.4.2): What was the interpolation method used in the DSM generation (TIN, bilineal, bicubic)?

Comment 4: P8 (section 2.5.2): How did you extracted the image coordinates? Could you detail the process (visually, number of iterations,...)?

Comment 5: P10 (section 3.3) and Discussion (section 4.1): The authors explain and numerically detail the accuracy of several positioning procedures, but it would be interesting to compare them with a standard (e.g. ASPRS http://www.asprs.org/a/society/divisions/pad/Accuracy/Comments_NGTOC_Rev5_V1.docx), especially regarding the vegetated and non-vegetated terrain.

Comment 6: P5 (section 2.3.1): Finally, the authors set the trigger interval in seconds, but they do not detail the rover velocity. Then, if the v is specified the reader could know how many meters lag between image captions and, if the GNSS rate is given, the distance between GNSS records.

Kind regards,

Please also note the supplement to this comment:
https://www.earth-surf-dynam-discuss.net/esurf-2019-2/esurf-2019-2-RC2-supplement.pdf

---

## Referee Comment (RC3) · Anonymous Referee #3 · 2 Apr 2019

The manuscript evaluated the repeatability of PPK UAV flight missions for precise topographic mapping. It is well structured and well written providing sufficient literature background and state-of-the-art methods. Results are presented from different perspectives and discussed broadly. The manuscript provides a contribution to the current debate of emerging PPK UAV data acquisition workflows that can be of interest to the readership of Earth Surface Dynamics. However, I encourage the authors to revise the manuscript based on some minor comments.

Scientific comments:

1) Compared to the entire manuscript, the introduction is very long, and some para-

graphs could be more concise as most of the text builds on the general knowledge base (e.g. general camera parameters, exterior orientation). Even though the authors stress the aims of the research, the novelty of this contribution is a bit fuzzy, as much research in this field has been done already. Efficiency is mentioned as one of the main objectives; however, there is little evidence on this in the results/discussion section as those parts mainly focus on repeatability/reproducibility.

2) Comparing metrical horizontal/vertical residuals of datasets with different GSD might not be the best approach, and normalized residuals could be more appropriate

3) Line 3 page 6: Can you provide more information on the decision not to use a cross-flight pattern or a single perpendicular strip as recommended by various authors?

4) Did your UAVs also record attitude parameters? In the manuscript, you should make clear why you used some observations of the external orientation parameters and others not.

5) You used Pix4D as a kind of black-box program. Which settings did you choose for camera calibration and the accuracy for geotagging information?

6) What are the reasons for the artificial quadratic pattern in the DoD of the DSLR in Figure 6a (upper left picture) – this almost looks like a kind of systematic error. This pattern needs to be explained in the text.

7) Some results are not very clear to the reader. I recommend extending the results section with some more explanation to enhance readability.

Technical corrections:

1) Figures: Check consistency of font and readability of legends

2) There are some typos and grammar mistakes – I recommend English proofreading prior to publication

**ESurfD**

---

## Author Comment (AC1) · 5 Apr 2019

**Point-by-point response Evaluating the Potential of PPK Direct Georeferencing for UAV-SfM Photogrammetry and Precise Topographic Mapping**

He Zhang, Emilien Aldana-Jague, François Clapuyt, Florian Wilken, Veerle Vanacker, Kristof Van Oost

5

We really appreciate the suggestions that can make our work more valuable. We gratefully thank our reviewers and the editor for their careful advices and changed the manuscript accordingly. Please see the detailed answers (in italics) to the comments below:

**Referee #1**

**10 General comments:**

The paper presents analyses of topographic data acquired by UAV and SfM-MVS photogrammetry using a PPK-GNSS direct georeferencing approach. This is a technique with broad relevance to a wide range of disciplines because the method will become increasingly widespread.

We welcome this assessment.

15

**Scientific comments:**

1) Nevertheless, novel findings within the work are difficult to identify clearly and I haven't found the methods section sufficiently detailed to fully understand what has been done. The contribution of the work would be much clearer if existing similar work was evaluated more critically to provide a detailed context, and the aims and outcomes more concisely defined.

- 20 Drawing more deeply on published work should allow statements of well-established principles (such as "camera properties have an impact on the accuracy") to be removed from the key sections such as abstract, discussion and conclusions so that the new findings can be more clearly communicated. The work is of interest but insufficiently described and, currently, the paper is somewhat challenging to assimilate. Overall, my suggestions below are aimed at highlighting the most transferrable new results from the work, by downplaying areas that have been previously covered and extending discussion to explore the
- 25 underlying concepts further.

We agree that a substantial rewrite of the paper following the suggestions made here, will improve its focus, readability and will also more clearly identify the novel aspects. We do think that an assessment of the repeatability, reproducibility and efficiency of a PPK-SfM workflow in the context of 4D earth surface monitoring with time-lapse SfM photogrammetry is timely and highly relevant for geomorphological research. In our answer to the detailed comments below, we discuss how these

30 *improvements can be implemented.*

2) With the paper focussed on PPK direct georeferencing for UAV surveys, the introduction would be well served by focussing on this. With UAV-SfM approaches not being so new, substantial regions of text (e.g. up to P 3), which introduce the broader aspects and uses of UAV photogrammetry could be condensed into a few sentences or a single paragraph. The introduction

5 would be strengthened by incorporating Table A1 into the main text and critically evaluating the progress of PPK controlled UAV surveys so far. Consideration of established use of this approach for crewed aircraft could be covered briefly. Inclusion of the recent PPP work by Grayson et al. (2018; DOI: 10.1111/phor.12259) – and references included within it – will also strengthen this section.

We will follow these suggestions. We will focus more on PPK direct georeferencing and remove the sections describing the
UAV-SfM approach. Table A1 will be presented and discussed in the introduction to improve the description of the state-of-the-art in relation to PPK and the problem statement. The PPP work by Grayson et al. (2018; DOI: 10.1111/phor.12259) will be included in Table A1 as well as a PPK work by Padró et al. (10.1016/j.jag.2018.10.018).

3) One aspect of the work is an exploration of predictors for survey repeatability. The rationale for some of the selections could

- 15 be strengthened here, and the utility of tie point density (which is shown to explain

5 Action camera:

Figure. Precision and error maps for simulated UAV surveys. The region encompassed by the dashed line is the same processing area of the propagated error maps based on the point density.

- 10 By comparing the two error maps derived from different approach, we observed highly consistent pattern of higher uncertainty/error area that can be distinguished from the DSLR camera dataset. Instead, the result of action camera showed no spatialized error pattern, which is also agree with the result that GoPro dataset was not spatially structured. To deeper interpret the Monte Carlo estimation, we calculated the XYZ precision (sXYZ) by points and then converted the points into raster (assigned by mean, resolution=0.92 m).
- 15

Precision estimated by Monte Carlo:

---

## Author Comment (AC2) · 5 Apr 2019

**Point-by-point response**
**Evaluating the Potential of PPK Direct Georeferencing for UAV-SfM Photogrammetry and Precise Topographic Mapping**

He Zhang, Emilien Aldana-Jague, François Clapuyt, Florian Wilken, Veerle Vanacker, Kristof Van Oost

*We really appreciate the suggestions that can make our work more valuable. We gratefully thank our reviewers and the editor for their careful advices and changed the manuscript accordingly. Please see the detailed answers (in italics) to the comments below:*

**Referee #1**

**General comments:**

The paper presents analyses of topographic data acquired by UAV and SfM-MVS photogrammetry using a PPK-GNSS direct georeferencing approach. This is a technique with broad relevance to a wide range of disciplines because the method will become increasingly widespread.

*We welcome this assessment.*

**Scientific comments:**

1) Nevertheless, novel findings within the work are difficult to identify clearly and I haven't found the methods section sufficiently detailed to fully understand what has been done. The contribution of the work would be much clearer if existing similar work was evaluated more critically to provide a detailed context, and the aims and outcomes more concisely defined. Drawing more deeply on published work should allow statements of well-established principles (such as "camera properties have an impact on the accuracy") to be removed from the key sections such as abstract, discussion and conclusions so that the new findings can be more clearly communicated. The work is of interest but insufficiently described and, currently, the paper is somewhat challenging to assimilate. Overall, my suggestions below are aimed at highlighting the most transferrable new results from the work, by downplaying areas that have been previously covered and extending discussion to explore the underlying concepts further.

*We agree that a substantial rewrite of the paper following the suggestions made here, will improve its focus, readability and will also more clearly identify the novel aspects. We do think that an assessment of the repeatability, reproducibility and efficiency of a PPK-SfM workflow in the context of 4D earth surface monitoring with time-lapse SfM photogrammetry is timely and highly relevant for geomorphological research. In our answer to the detailed comments below, we discuss how these improvements can be implemented.*

2) With the paper focussed on PPK direct georeferencing for UAV surveys, the introduction would be well served by focussing on this. With UAV-SfM approaches not being so new, substantial regions of text (e.g. up to P 3), which introduce the broader aspects and uses of UAV photogrammetry could be condensed into a few sentences or a single paragraph. The introduction would be strengthened by incorporating Table A1 into the main text and critically evaluating the progress of PPK controlled UAV surveys so far. Consideration of established use of this approach for crewed aircraft could be covered briefly. Inclusion of the recent PPP work by Grayson et al. (2018; DOI: 10.1111/phor.12259) – and references included within it – will also strengthen this section.

*We will follow these suggestions. We will focus more on PPK direct georeferencing and remove the sections describing the UAV-SfM approach. Table A1 will be presented and discussed in the introduction to improve the description of the state-of-the-art in relation to PPK and the problem statement. The PPP work by Grayson et al. (2018; DOI: 10.1111/phor.12259) will be included in Table A1 as well as a PPK work by Padró et al. (10.1016/j.jag.2018.10.018).*

3) One aspect of the work is an exploration of predictors for survey repeatability. The rationale for some of the selections could be strengthened here, and the utility of tie point density (which is shown to explain <50% of the variance for one camera) more critically considered. How useful is this, given that the analysis only appears to work for one camera and requires deployment of GCPs to determine the relationship? The number of tie points retained per image is usually a software setting that can be varied. Consequently, any parameterisation of repeatability would be software and UAV system dependent. Furthermore, other more important parameters are not considered. Within the bundle adjustment, measurement precision for a tie point is related to the number and angles of observations – how do these vary? The authors cite James et al. (2017) who show how point coordinate precision varies spatially and can be linked directly to these photogrammetric factors and other georeferencing factors within the adjustment. Consequently, maps of 3D point precision can be determined without any GCPs. The work here would be strengthened by discussing the authors 'spatialised error' approach in context with the 3D precision maps of James et al. (2017). The authors could also consider the findings of Mosbrucker et al (2017; DOI: 10.1002/esp.4066) within the discussion (or introduction).

*The section were we presented a relation between tie point density and precision simply showed that it is a good indicator which can also be used to estimate the error patterns. We are aware of the fact that different sensors/flight patterns/surfaces/weather/illumination conditions could lead to differences in tie point magnitude. We agree with the author that the relationship requires deployment of GCPs and is empirical. However, we show below that the relationship is robust (see figure below) and could be used in temporal monitoring.*

*We suggest to substantially expand this section on tie point uncertainties by implementing the approach presented by James et al 2017 (based on monte carlo simulation) and by comparing this with our independent quantification of precision using GCP-based precision assessments. Following the workflow by James et al., 2017, we analysed DSLR camera (EOS) and action camera (GoPro) datasets, respectively. During the alignment process, georeferencing was achieved by PPK coordinates*

*without GCP reference. The Monte Carlo processing comprised 4000 iterations for each survey, and tie point uncertainties were calculated in X, Y and Z directions. The results are as follows:*

*DSLR camera:*

[Figure]

5    *Action camera:*

[Figure]

*Figure. Precision and error maps for simulated UAV surveys. The region encompassed by the dashed line is the same processing area of the propagated error maps based on tie point density.*

10    *By comparing the two error maps derived from different approach, we observed highly consistent pattern of higher uncertainty/error area that can be distinguished from the DSLR camera dataset. Instead, the result of action camera showed no spatialized error pattern, which is also agree with the result that GoPro dataset was not spatially structured. To deeper interpret the Monte Carlo estimation, we calculated the XYZ precision (sXYZ) by points and then converted the points into raster (assigned by mean, resolution=0.92 m).*

15

*Precision estimated by Monte Carlo:*

[Figure]

*Precision estimated by tie point density:*

[Figure]

*Figure. Precision map estimated based on Monte Carlo and tie point density. The XYZ error was estimated by the density-error regression model (y = -0.00041x + 0.125)*

*We observed similar shapes in vegetation area with higher error and the "artificial quadratic pattern" with less error in overlapping area between adjacent images. To test the consistency of the two estimation approaches, we randomly sampled 500 points within the region of interest and extracted values from the two maps, the results were as follows:*

[Figure]

*To better represent the real variability, we used Latin Hypercube sampling and selected 20 points to compare the two estimated datasets. The results were as follows:*

[Figure]

5    *Therefore, the estimation based on tie point density could be a quick alternative approach of evaluating tie point precision and spatializing the error.*

4) Comparison of results from different cameras (S4.2; particularly the last paragraph) dominantly reflects established relationships between camera/flight parameters and conventional survey design principles. In my view, this material should

10   form the rationale behind the survey design, and be given within the introduction or methods sections. Placing this within the discussion detracts from the newer aspects of the work (the PPK).

*We will adjust the manuscript accordingly.*

5) Throughout, when discussing results from different cameras, I suggest that a dimensionless approach based on pixels (or ground sampling distance) is also used. This could be used to assess the quality of the photogrammetric networks achieved, and to generate insight – again, see previous work, including that of Mosbrucker et al. (2017). I would actually see a much more detailed assessment of the PPK performance as providing the most useful (i.e. transferrable) insight.

5 *Thank you! We will add the calculation of relative error based on GSD and the distance between surface and camera.*

*The relative error is defined as the ratio between measured RMSE and surface to camera distance: $e_r = \frac{\sigma_m}{D}$, where $e_r$ is the relative error, $\sigma_m$ the measured RMSE and D the mean distance between the camera and surface.*

*Table 2 will be extended accordingly:*

| Camera | Flight Mission Date | Flight Height (m) | Mean(m) | | | | RMSE(m) | | | | Relative Error (px/m) | |
|---|---|---|---|---|---|---|---|---|---|---|---|---|
| | | | X | Y | XY | Z | X | Y | XY | Z | XY | Z |
| DSLR camera (EOS) | 21, March | 45 | 0.011 | -0.011 | 0.016 | 0.012 | 0.016 | 0.015 | 0.022 | 0.017 | 0.071 | 0.055 |
| | 30, March | 45 | 0.009 | 0.013 | 0.015 | 0.016 | 0.028 | 0.021 | 0.035 | 0.027 | 0.113 | 0.087 |
| | 05, April | 45 | 0.005 | 0.014 | 0.014 | 0.003 | 0.014 | 0.02 | 0.024 | 0.022 | 0.077 | 0.071 |
| | 06, April | 35 | 0.006 | 0 | 0.006 | 0.019 | 0.008 | 0.004 | 0.008 | 0.023 | 0.042 | 0.122 |
| Action camera (GoPro) | 21, March | 45 | 0.018 | 0.021 | 0.027 | 0.049 | 0.028 | 0.031 | 0.042 | 0.076 | 0.039 | 0.071 |
| | 30, March | 45 | 0.008 | 0.008 | 0.011 | -0.042 | 0.016 | 0.015 | 0.022 | 0.051 | 0.021 | 0.048 |
| | 05, April | 45 | -0.01 | 0.008 | 0.013 | 0.009 | 0.014 | 0.013 | 0.019 | 0.024 | 0.018 | 0.022 |
| | 06, April | 20 | -0.013 | 0.02 | 0.024 | -0.015 | 0.018 | 0.024 | 0.03 | 0.02 | 0.129 | 0.086 |

6) It would be good to see some more details to support the photogrammetric processing though – e.g. what were the rms image residual magnitudes? Did they vary image-to-image in any way that would help understanding of the repeatability? The clear image overlap outlines shown in Fig 6a suggest that camera positions may have been over-constrained in at least one survey (e.g. see a similar effect in Fig 1 of James et al. 2017a - http://dx.doi.org/10.1016/j.geomorph.2016.11.021, resulting

15 from overweighting the GCPs in that case). Details of the a priori assigned camera position precisions used in the adjustments need to be provided and, given that they are often optimistically estimated, the effects of diluting the estimates could be explored.

*We will add these analyses, including the effect of diluting the camera position precisions.*

*As for the image residuals, we selected two datasets from DSLR and action camera and depicted the planimetric residuals*

20 *between calibrated position and original position by Pix4D SfM processing. The results are as follows:*

*DSLR camera:*

[Figure]

*Action camera:*

[Figure]

*Figure. Residuals on the images and CPs in planimetric view (vectors give the horizontal residual component magnified by ×500 and ×100 for DSLR and action camera, respectively).*

*For the DSLR camera residuals, the outliers were mainly related to the points where the drone turned, while the rest were mostly within 1 cm (which is consistent with the RTK GPS precision). For the action camera dataset, the image residuals were larger while the CP residuals were not higher than those for the DSLR dataset. We ascribe this difference to the compensation of more convergent images from wider angle as discussed in the manuscript.*

*The clear overlap outlines (Fig 6a) were observed for the low FOV (field of view) camera. With a diagonal FOV of ca. 43°, one tie point can be observed simultaneously by 7-12 images for EOS camera in our case, while this is 60-80 images for the GoPro camera. This had an effect on the BBA process, where the GoPro output had no such overlap outline effects. Though the image overlap outlines shift in our case is ca. 2 cm, which can be eliminated by applying DoD threshold, we will investigate how image constraint rigidity impacts on the output accuracy as well as the quadratic outline pattern.*

| camera | Vertical FOV | Horizontal FOV | Diagonal FOV |
|---|---|---|---|
| EOS
Canon APS-C, lens 35 mm | 24° | 36° | 43° |
| GoPro
4×3 Wide (Zoom = 0%) | 94.4° | 122.6° | 149.2° |

Fig. 2 Are photographs of the GPS system etc. really needed (c, d)? Much more valuable would be examples of the imagery processed (i.e. the underpinning data on which the work relies), with enlarged excerpts to illustrate the image quality and show how the GCPs have been imaged.

*We will modify this figure and add the image-vision of GCPs.*

All figures need to be checked for readability of the text labels. In particular, all map figures have scale and other labels which are far too small to be readable, and font sizes vary substantially across the figures. Labels must be readable: more consistent font sizes, of at least 9 point, will help.

*Thank you for this comment, we will follow the suggestion.*

Fig 4 Rephrase 'error of detection' for clarity.

DoDs – represent image overlaps etc for the DSLR but not for the action camera.

*We will rephrase this concept.*

Fig 7b LoD before/after lines indistinguishable – needs more careful visualisation.

*We will follow the suggestion.*

Fig. 8 Colour scale given to four decimal places could be tidied up.

*We will follow the suggestion.*

5    Table 1 The caption mentions three flights but I can only see data from two (i.e. one with DLSR, one with action camera). Which flights are these? Where are the results from the others? Table 2 The link to Table 1 is unclear. 05 April DLSR results are the same as in Table 1, but no similar repetition for Action camera. Maybe I haven't understood what Table 1 is?

*We will add the information from all the flights conducted (in correspondence with data presented in Table 2).*

10    Fig A1. I am not convinced how useful these visualisations are – it is difficult to extract much from them. I would suggest that a more informative plot would be as an XY map of discrepancy vectors, with symbols to indicate the check point position and Z-discrepancy showed by symbol colour. This way, any spatial systematics (which would be concealed in the current plots) would be clear.

*We will follow the suggestion.*

15

Table A2 This information is critical to a reader's understanding; it needs to be early in the main manuscript, not in the appendices. Why were some flights repeated? Where are these repeated data, and what did they show?

*We will follow the suggestion.*

20

**Referee #2**

**General comments:**

The paper "Evaluating the Potential of PPK direct Georeferencing for UAV-SfM Photogrammetry and Precise Topographic
Mapping" fits the scope of the journal and I consider that the paper is very interesting for the Earth Surface Dynamics'
readership. Moreover, it is a well-written paper, with very interesting results and rigorous validations. However, some minor
revisions and comments must be fixed before the final publication:

*Below, we show how we have revised the manuscript in light of these comments and recommendations.*

**Scientific comments:**

1) Introduction (section 1) and Discussion (section 4.2): There is a very recent publication where it is compared the accuracy
of different PPK approaches and other positioning alternatives, using DLSR cameras (10.1016/j.jag.2018.10.018). This could
be in the introduction and in the discussion, since this research follows a similar workflow.

*We will follow the suggestion and add this work by Padró et al. (10.1016/j.jag.2018.10.018). We will also include the PPP*
*work by Grayson et al. (2018; DOI: 10.1111/phor.12259) in Table A1 to strengthen the advance in direct georeferencing.*

2) P6 (section 2.3.2): Why did you not post-processed the static GNSS measurements?

*The positioning measurement of GCPs was conducted using Reach RS in RTK mode, for which the differential correction data*
*was transmitted via mobile IP network (the same approach for determining the base station coordinate). The Reach RS + GPS*
*pole setup could provide a better control of antenna placement and reduce disturbance. We also tested the precision of the*
*RTK solution by repeatedly measuring a fixed point near our department (12 km from the study field and 20 km from the BRUS*
*station), and the error was ca. 0.010 m in XYZ. We thereby determined GCPs by RTK solution.*

3) P7 (section 2.4.2): What was the interpolation method used in the DSM generation (TIN, bilinear, bicubic)?

*It is the mean altitude from the point cloud data which is assigned to the DSM raster values. We will add this information in*
*the revised manuscript.*

4) P8 (section 2.5.2): How did you extracted the image coordinates? Could you detail the process (visually, number of
iterations, . . .)?

*After initial processing of pix4d, the software will generate a file with optimized position information based on the distortion*
*model. The external camera parameters are given by:*

*T = (Tx, Ty, Tz) the position of the camera projection center in world coordinate system.*

*R the rotation matrix that defines the camera orientation with angles ω, φ, κ (PATB convention.)*

*If X = (X, Y, Z) is a 3D point in world coordinate system, its position X' = (X', Y', Z') in camera coordinate system is given by:*

$$X' = R^T(X - T)$$

*This was automatically computed in the initial processing, and the external camera parameters can be derived from the output "txt." file.*

5) P10 (section 3.3) and Discussion (section 4.1): The authors explain and numerically detail the accuracy of several positioning procedures, but it would be interesting to compare them with a standard (e.g. ASPRS [http://www.asprs.org/a/society/divisions/pad/Accuracy/Comments_NGTOC_Rev5_V1.docx](http://www.asprs.org/a/society/divisions/pad/Accuracy/Comments_NGTOC_Rev5_V1.docx)), especially regarding the vegetated and non-vegetated terrain.

10    *Thank you! We will strengthen this section.*

6) P5 (section 2.3.1): Finally, the authors set the trigger interval in seconds, but they do not detail the rover velocity. Then, if the v is specified the reader could know how many meters lag between image captions and, if the GNSS rate is given,the distance between GNSS records.

15    *The rover velocity is determined when the front-overlap, flight height, trigger interval and the camera parameter (especially horizontal FOV) are preset. Given that the trigger interval for DSLR camera (EOS) was 2 s and action camera (GoPro) was 4 s, the velocity was ca. 4.5 m/s and 2.8 m/s, respectively. The corresponding distance between each capture were 9 m for EOS and 11 m for GoPro. The image information can also be derived from the GPS logging or Pix4D output log after initial post-processing.*

20

**Referee #3**

**General comments:**

The manuscript evaluated the repeatability of PPK UAV flight missions for precise topographic mapping. It is well structured and well written providing sufficient literature background and state-of-the-art methods. Results are presented from different perspectives and discussed broadly. The manuscript provides a contribution to the current debate of emerging PPK UAV data acquisition workflows that can be of interest to the readership of Earth Surface Dynamics. However, I encourage the authors to revise the manuscript based on some minor comments.

**Scientific comments:**

1) Compared to the entire manuscript, the introduction is very long, and some paragraphs could be more concise as most of the text builds on the general knowledgebase (e.g. general camera parameters, exterior orientation). Even though the authors stress the aims of the research, the novelty of this contribution is a bit fuzzy, as much research in this field has been done already. Efficiency is mentioned as one of the main objectives; however, there is little evidence on this in the results/discussion section as those parts mainly focus on repeatability/reproducibility.

*We will compact the introduction and improve its focus, readability and will also identify the novel aspects more clearly (see also above).*

2) Comparing metrical horizontal/vertical residuals of datasets with different GSD might not be the best approach, and normalized residuals could be more appropriate.

*We will follow the suggestion and Referee #1 has similar comment to use a dimensionless approach. We will add the calculation of relative error based on GSD and the distance between surface and camera.*

*"The relative error is defined as the ratio between measured RMSE and surface to camera distance: $e_r = \frac{\sigma_m}{D}$, where $e_r$ is the relative error, $\sigma_m$ the measured RMSE and D the mean distance between the camera and surface."*

*Table 2 will be extended accordingly:*

| Camera | Flight Mission Date | Flight Height (m) | Mean(m) | | | | RMSE(m) | | | | Relative Error (px/m) | |
|---|---|---|---|---|---|---|---|---|---|---|---|---|
| | | | X | Y | XY | Z | X | Y | XY | Z | XY | Z |
| DSLR camera (EOS) | 21, March | 45 | 0.011 | -0.011 | 0.016 | 0.012 | 0.016 | 0.015 | 0.022 | 0.017 | 0.071 | 0.055 |
| | 30, March | 45 | 0.009 | 0.013 | 0.015 | 0.016 | 0.028 | 0.021 | 0.035 | 0.027 | 0.113 | 0.087 |
| | 05, April | 45 | 0.005 | 0.014 | 0.014 | 0.003 | 0.014 | 0.02 | 0.024 | 0.022 | 0.077 | 0.071 |
| | 06, April | 35 | 0.006 | 0 | 0.006 | 0.019 | 0.008 | 0.004 | 0.008 | 0.023 | 0.042 | 0.122 |

| Action camera (GoPro) | 21, March | 45 | 0.018 | 0.021 | 0.027 | 0.049 | 0.028 | 0.031 | 0.042 | 0.076 | 0.039 | 0.071 |
| | 30, March | 45 | 0.008 | 0.008 | 0.011 | -0.042 | 0.016 | 0.015 | 0.022 | 0.051 | 0.021 | 0.048 |
| | 05, April | 45 | -0.01 | 0.008 | 0.013 | 0.009 | 0.014 | 0.013 | 0.019 | 0.024 | 0.018 | 0.022 |
| | 06, April | 20 | -0.013 | 0.02 | 0.024 | -0.015 | 0.018 | 0.024 | 0.03 | 0.02 | 0.129 | 0.086 |

3) Line 3 page 6: Can you provide more information on the decision not to use a crossflight pattern or a single perpendicular strip as recommended by various authors?

*One key aspect of our study is to provide practical tips for UAV survey, thus the autonomy (flight duration) is an important factor to take into account. A crosshatch pattern could improve accuracy by providing multi-angle images and increased overlap. However, it may double the survey time and battery consumption, and the promotion for accuracy is limited since the PPK enabled precise image georeferencing. To this end, we considered the tradeoff between survey accuracy and efficiency and opted the non-cross pattern throughout the experiment. It is also our motivation to investigate the large-FOV (field of view) action camera, so as to find a better solution for accuracy/efficiency ratio.*

4) Did your UAVs also record attitude parameters? In the manuscript, you should make clear why you used some observations of the external orientation parameters and others not.

*Sorry we didn't make this clear. Our cameras used a separated system without connecting to IMU, so the images only contained positional information without attitude parameters. We will rephrase the method section to make it clear.*

5) You used Pix4D as a kind of black-box program. Which settings did you choose for camera calibration and the accuracy for geotagging information?

*We will add details in 2.4.2 Point Cloud and DSM Generation. "The horizontal and vertical accuracy were both set to 0.05 m. We kept the remaining settings as default as 3D maps template, i.e., full keypoints image scale, automatic targeted number of keypoints and standard calibration method."*

6) What are the reasons for the artificial quadratic pattern in the DoD of the DSLR in Figure 6a (upper left picture) – this almost looks like a kind of systematic error. This pattern needs to be explained in the text.

*The artificial quadratic pattern of the DSLR camera were usually observed especially for the low-FOV camera. With a diagonal FOV ca. 43°, one tie point can be observed simultaneously by 7-12 images for EOS camera in our case. While we compared the output with the large-FOV camera GoPro and one single point can be captured by 60-80 images from wider angle. This had different effects on the BBA process, e.g., GoPro output had no such outline effects. Though the image overlap outlines shift in our case is ca. 2 cm, which can be eliminated by applying DoD threshold, we will investigate how image constraint rigidity impacts on the output accuracy as well as the quadratic outline pattern.*

7) Some results are not very clear to the reader. I recommend extending the results section with some more explanation to enhance readability.

*We will follow this suggestion.*

**Technical corrections:**

1) Figures: Check consistency of font and readability of legends

*We will follow the suggestion.*

2) There are some typos and grammar mistakes – I recommend English proofreading prior to publication.

10    *We will carefully correct these.*

---

## Author Response (AR1)

**Point-by-point response**
**Evaluating the Potential of PPK Direct Georeferencing for UAV-SfM Photogrammetry and Precise Topographic Mapping**

He Zhang, Emilien Aldana-Jague, François Clapuyt, Florian Wilken, Veerle Vanacker, Kristof Van Oost

We thank the reviewers and the editor for their careful reviews. We considered their advice and changed the manuscript accordingly. In our response below, referee comments are shown in *blue*, our response in **black** and changes in the manuscript in red. We have also attached a revised manuscript with highlighted changes: page numbers in the text below refer to this highlighted version of the manuscript.

Thank you for considering our revised manuscript.

Sincerely,

The authors

**Referee #1**

**General comments:**

*The paper presents analyses of topographic data acquired by UAV and SfM-MVS photogrammetry using a PPK-GNSS direct georeferencing approach. This is a technique with broad relevance to a wide range of disciplines because the method will become increasingly widespread.*

We welcome this assessment.

**Scientific comments:**

*1) Nevertheless, novel findings within the work are difficult to identify clearly and I haven't found the methods section sufficiently detailed to fully understand what has been done. The contribution of the work would be much clearer if existing similar work was evaluated more critically to provide a detailed context, and the aims and outcomes more concisely defined.*

*Drawing more deeply on published work should allow statements of well-established principles (such as "camera properties have an impact on the accuracy") to be removed from the key sections such as abstract, discussion and conclusions so that the new findings can be more clearly communicated. The work is of interest but insufficiently described and, currently, the paper is somewhat challenging to assimilate. Overall, my suggestions below are aimed at highlighting the most transferrable new results from the work, by downplaying areas that have been previously covered and extending discussion to explore the underlying concepts further.*

We do think that an assessment of the repeatability, reproducibility and efficiency of a PPK-SfM workflow in the context of 4D earth surface monitoring with time-lapse SfM photogrammetry is timely and highly relevant for geomorphological research. There is some recent research on the accuracy of PPK direct georeferencing but not in the context of 4D monitoring. The consistency of data generation over longer monitoring periods and how uncertainty is propagated is crucial for geomorphological applications. This is the main focus of this paper. In addition, we would like to emphasize that we do not repeat 'well-established principles in relation to camera properties. Rather we focus on two typical UAV-camera setups: We are convinced that an evaluation of the performance of a micro-drone equipped with an action camera (low-cost and very high portability) versus a heavy professional UAV equipped with a reflex camera is relevant for the readership of your journal. Based on these comments, we have substantially rewritten the introduction and method sections following the suggestions made here. We have also addressed the issues of novelty and transferability. In our answers to the detailed comments below, we discuss how these improvements were implemented.

*2) With the paper focussed on PPK direct georeferencing for UAV surveys, the introduction would be well served by focussing on this. With UAV-SfM approaches not being so new, substantial regions of text (e.g. up to P 3), which introduce the broader aspects and uses of UAV photogrammetry could be condensed into a few sentences or a single paragraph. The introduction would be strengthened by incorporating Table A1 into the main text and critically evaluating the progress of PPK controlled UAV surveys so far. Consideration of established use of this approach for crewed aircraft could be covered briefly. Inclusion of the recent PPP work by Grayson et al. (2018; DOI: 10.1111/phor.12259) – and references included within it – will also strengthen this section.*

We have rewritten the introduction section and focused more on PPK direct georeferencing. We also removed the introduction of other DEM acquisition methods. Table A1 was presented and discussed in the introduction as Table 1 to improve the description of the state-of-the-art in relation to PPK and the problem statement. The PPP work by Grayson et al. (2018; DOI: 10.1111/phor.12259) was included in Table 1 as well as a PPK work by Padró et al. (10.1016/j.jag.2018.10.018).

*3) One aspect of the work is an exploration of predictors for survey repeatability. The rationale for some of the selections could be strengthened here, and the utility of tie point density (which is shown to explain <50% of the variance for one camera) more critically considered. How useful is this, given that the analysis only appears to work for one camera and requires deployment of GCPs to determine the relationship? The number of tie points retained per image is usually a software setting that can be varied. Consequently, any parameterisation of repeatability would be software and UAV system dependent. Furthermore, other more important parameters are not considered. Within the bundle adjustment, measurement precision for a tie point is related to the number and angles of observations – how do these vary? The authors cite James et al. (2017) who show how point coordinate precision varies spatially and can be linked directly to these photogrammetric factors and other georeferencing factors within the adjustment. Consequently, maps of 3D point precision can be determined without any*

*GCPs. The work here would be strengthened by discussing the authors 'spatialised error' approach in context with the 3D precision maps of James et al. (2017). The authors could also consider the findings of Mosbrucker et al (2017; DOI: 10.1002/esp.4066) within the discussion (or introduction).*

We have completely reworked this section and we now use two approaches to estimate precision for the change detection analysis: observational (i.e. CP-derived) precision and Monte Carlo-derived precision (using the workflow proposed by James et al 2017). The analysis are presented in Fig. 5 and Fig. A3 and are discussed on page 18.

*4) Comparison of results from different cameras (S4.2; particularly the last paragraph) dominantly reflects established relationships between camera/flight parameters and conventional survey design principles. In my view, this material should form the rationale behind the survey design, and be given within the introduction or methods sections. Placing this within the discussion detracts from the newer aspects of the work (the PPK).*

We have reduced these contents and discussed more about how camera properties impact on SfM output precision and accuracy. We retained some discussion about the low FOV feature. It not only determines the survey design, but also affects the tie point matching due to a more intense radial distortion, which affects the accuracy.

To deeper understand what difference comes out from the low FOV feature regarding to tie point precision, we carried out some additional flight survey using the low FOV action camera. In this survey, we used a denser flight plan to have a higher overlap and we clipped the images to retain the central area of each image. The new dataset has a much lower radial distortion. We also down-sampled the resolution of the DSLR dataset to have the same GSD as the action camera images, this allowed us to evaluate the variate (GSD or FOV) controlling tie point uncertainties (P19).

Both precision map exhibited spatialized pattern, we thereby infer that the low FOV resulted in the non-spatialized precision of the action camera dataset. It also indicated that the low FOV affected tie point matching. With a diagonal FOV of ca. 43º, one tie point can be observed simultaneously by 7-12 images for DSLR camera in our case, while 60-80 images for the action camera which are from wider imaging angles.

| Camera | Vertical FOV | Horizontal FOV | Diagonal FOV |
|---|---|---|---|
| DSLR camera: EOS Canon APS-C, lens 35 mm | 24º | 36º | 43º |
| Action camera: GoPro 4×3 Wide (Zoom = 0%) | 94.4º | 122.6º | 149.2º |

*5) Throughout, when discussing results from different cameras, I suggest that a dimensionless approach based on pixels (or ground sampling distance) is also used. This could be used to assess the quality of the photogrammetric networks achieved, and to generate insight – again, see previous work, including that of Mosbrucker et al. (2017). I would actually see a much more detailed assessment of the PPK performance as providing the most useful (i.e. transferrable) insight.*

We presented the accuracy results of each survey using density plots, showing the CP residuals on X, Y and Z directions. The units were given in both meters and pixels. The pixels were calculated based on GSD at corresponding flight height. We expressed the accuracies in pixels to standardize the RMSEs in terms of the expected error incurred from GSD. The CP XYZ RMSEs for the DSLR camera corresponds to ca. 4-15 pixels. However, it should be noted that the GSD for the DSLR camera

5    is extremely fine (0.006 m) due to the low flight height, and this is much finer than the width of the markers used on the CP (0.02 m) or the precision of the CPs. As a result, the XYZ RMSEs for the action camera were better and within a range of 1 to 5 pixels. The extreme high GSD (down to 6 mm) makes it difficult to compare it to reported accuracies in terms of pixels. However, by reporting both pixels and meters, and by discussing the specific conditions of our analysis, we hope we have sufficiently addressed the issue of transferability.

*6) It would be good to see some more details to support the photogrammetric processing though – e.g. what were the rms image residual magnitudes? Did they vary image-to-image in any way that would help understanding of the repeatability? The clear image overlap outlines shown in Fig 6a suggest that camera positions may have been over-constrained in at least one survey (e.g. see a similar effect in Fig 1 of James et al. 2017a - http://dx.doi.org/10.1016/j.geomorph.2016.11.021,*

15    *resulting from overweighting the GCPs in that case). Details of the a priori assigned camera position precisions used in the adjustments need to be provided and, given that they are often optimistically estimated, the effects of diluting the estimates could be explored.*

We added the images residuals plot in the appendix, showing the shift between the original image positions and the optimized position after BBA process. The DSLR images had a smaller SD than the action camera images, indicating that the action

20    camera images had higher random error regarding the BBA process.

[Figure]

[Figure]

**Figure. Residuals on the images and CPs in planimetric view. Vectors give the horizontal residual component magnified by ×500 for DSLR survey (left) and ×100 for action camera survey(right). With inset mean value and standard deviation of the image residuals.**

As for the image overlap outlines of the DSLR datasets, we agreed that the camera positions might have been over-constrained. However, the shift between areas where constructed from the two adjacent images was less than 2 cm, which can be eliminated by applying a LoD threshold. In order to find a balance between the camera accuracy (BBA liberty) and the tie point reliability for the *PPK* solution, we implemented a test of setting different camera accuracy (from 0.01 m to 1 m) and validated the precision by CP residuals and corresponding Monte Carlo extracted values. We found a strong relation between precision and camera accuracy. It indicated that for direct georeferencing, the precision of tie points was determined by camera accuracy, which means that the camera accuracy should be wisely set. It should be noted that the overlap outlines were only observed in the low FOV DSLR dataset, whereas the fisheye action camera output had no such outline effect.

[Figure]

**Figure. The relationship between precision and camera accuracy.**

We also explored the relation between tie point uncertainty and tie point density. From the figure we inferred that denser tie points generally lead to robust positioning of tie points. It implies that areas with relatively higher tie point density are of higher probability to be accurate and precise in the SfM output. With increased survey (images) density, the accuracy can be further improved when using RTK/PPK precise positioning, before which photogrammetric considerations are the limiting factors.

[Figure]

[Figure]

**Figure. (a) Tie Point density and Monte Carlo precision map. Dataset: F2a of DSLR survey. (b) Monte Carlo precision and tie point density for each cell of the map.**

5    *Fig. 2 Are photographs of the GPS system etc. really needed (c, d)? Much more valuable would be examples of the imagery processed (i.e. the underpinning data on which the work relies), with enlarged excerpts to illustrate the image quality and show how the GCPs have been imaged.*

We have modified this figure and add an image of GCPs.

10   *All figures need to be checked for readability of the text labels. In particular, all map figures have scale and other labels which are far too small to be readable, and font sizes vary substantially across the figures. Labels must be readable: more consistent font sizes, of at least 9 point, will help.*

We have adjusted the size of figures to make it more readable.

15   *Fig 4 Rephrase 'error of detection' for clarity.*

*DoDs – represent image overlaps etc for the DSLR but not for the action camera.*

We have rephrased the caption, i.e., spatialized error threshold for DSLR camera and constant error threshold for action camera.

*Fig 7b LoD before/after lines indistinguishable – needs more careful visualisation.*

5    We have adjusted the size.

*Fig. 8 Colour scale given to four decimal places could be tidied up.*

We have modified the figure to have the same decimal.

10    *Table 1 The caption mentions three flights but I can only see data from two (i.e. one with DLSR, one with action camera). Which flights are these? Where are the results from the others? Table 2 The link to Table 1 is unclear. 05 April DLSR results are the same as in Table 1, but no similar repetition for Action camera. Maybe I haven't understood what Table 1 is?*

We have checked the data and re-calculated the Mean, SD and RMSE error for all the surveys (in correspondence with data presented in Table 2).

*Fig A1. I am not convinced how useful these visualisations are – it is difficult to extract much from them. I would suggest that a more informative plot would be as an XY map of discrepancy vectors, with symbols to indicate the check point position and Z-discrepancy showed by symbol colour. This way, any spatial systematics (which would be concealed in the current plots) would be clear.*

20    We have removed this figure.

*Table A2 This information is critical to a reader's understanding; it needs to be early in the main manuscript, not in the appendices. Why were some flights repeated? Where are these repeated data, and what did they show?*

We have followed the suggestion. Now it is displayed as Table 2 in the main manuscript.

**Referee #2**

**General comments:**

*The paper "Evaluating the Potential of PPK direct Georeferencing for UAV-SfM Photogrammetry and Precise Topographic Mapping" fits the scope of the journal and I consider that the paper is very interesting for the Earth Surface Dynamics' readership. Moreover, it is a well-written paper, with very interesting results and rigorous validations. However, some minor revisions and comments must be fixed before the final publication:*

Below, we show how we have revised the manuscript in light of these comments and recommendations.

**Scientific comments:**

*1) Introduction (section 1) and Discussion (section 4.2): There is a very recent publication where it is compared the accuracy of different PPK approaches and other positioning alternatives, using DLSR cameras (10.1016/j.jag.2018.10.018). This could be in the introduction and in the discussion, since this research follows a similar workflow.*

We have followed the suggestion and added the work by Padró et al. (10.1016/j.jag.2018.10.018). We will also include the PPP work by Grayson et al. (2018; DOI: 10.1111/phor.12259) in Table 1 to strengthen the advance in direct georeferencing.

*2) P6 (section 2.3.2): Why did you not post-processed the static GNSS measurements?*

The positioning measurement of GCPs was conducted using Reach RS in RTK mode, for which the differential correction data was transmitted via mobile IP network (the same approach for determining the base station coordinate). The Reach RS + GPS pole setup could provide a better control of antenna placement and reduce disturbance. We also tested the precision of the RTK solution by repeatedly measuring a fixed point near our department (12 km from the study field and 20 km from the BRUS station), and the error was ca. 0.010 m in XYZ. We thereby determined GCPs by RTK solution.

*3) P7 (section 2.4.2): What was the interpolation method used in the DSM generation (TIN, bilinear, bicubic)?*

It is the mean altitude from the point cloud data which is assigned to the DSM raster values. We added this information in the revised manuscript (P9L2-3).

*4) P8 (section 2.5.2): How did you extracted the image coordinates? Could you detail the process (visually, number of iterations, . . .)?*

After initial processing of pix4d, the software will generate a file with optimized position information based on the distortion model. The external camera parameters are given by:

T = (Tx, Ty, Tz) the position of the camera projection center in world coordinate system.

R the rotation matrix that defines the camera orientation with angles ω, φ, κ (PATB convention.)

If X = (X, Y, Z) is a 3D point in world coordinate system, its position X' = (X', Y', Z') in camera coordinate system is given by:

$$X' = R^T(X - T)$$

This was automatically computed in the initial processing, and the external camera parameters can be derived from the output "txt." file.

We have substantially rewritten the section to include the detailed approach for obtaining precision map (section 2.5.2-2.5.3).

*5) P10 (section 3.3) and Discussion (section 4.1): The authors explain and numerically detail the accuracy of several positioning procedures, but it would be interesting to compare them with a standard (e.g. ASPRS http://www.asprs.org/a/society/divisions/pad/Accuracy/Comments_NGTOC_Rev5_V1.docx), especially regarding the vegetated and non-vegetated terrain.*

We have strengthened this discussion (section 4.2).

*6) P5 (section 2.3.1): Finally, the authors set the trigger interval in seconds, but they do not detail the rover velocity. Then, if the v is specified the reader could know how many meters lag between image captions and, if the GNSS rate is given, the distance between GNSS records.*

The rover velocity is determined when the front-overlap, flight height, trigger interval and the camera parameter (especially horizontal FOV) are preset. Given that the trigger interval for DSLR camera (EOS) was 2 s and action camera (GoPro) was 4 s, the velocity was ca. 4.5 m/s and 2.8 m/s, respectively. The corresponding distance between each capture were 9 m for EOS and 11 m for GoPro. The image information can also be derived from the GPS logging or Pix4D output log after initial post-processing.

**Referee #3**

**General comments:**

*The manuscript evaluated the repeatability of PPK UAV flight missions for precise topographic mapping. It is well structured and well written providing sufficient literature background and state-of-the-art methods. Results are presented from different perspectives and discussed broadly. The manuscript provides a contribution to the current debate of emerging PPK UAV data acquisition workflows that can be of interest to the readership of Earth Surface Dynamics. However, I encourage the authors to revise the manuscript based on some minor comments.*

**Scientific comments:**

*1) Compared to the entire manuscript, the introduction is very long, and some paragraphs could be more concise as most of the text builds on the general knowledgebase (e.g. general camera parameters, exterior orientation). Even though the authors stress the aims of the research, the novelty of this contribution is a bit fuzzy, as much research in this field has been done already. Efficiency is mentioned as one of the main objectives; however, there is little evidence on this in the results/discussion section as those parts mainly focus on repeatability/reproducibility.*

We have compacted the introduction and improved its focus, readability (section 1).

*2) Comparing metrical horizontal/vertical residuals of datasets with different GSD might not be the best approach, and normalized residuals could be more appropriate.*

We will follow the suggestion and Referee #1 has similar comment to use a dimensionless approach. We have reorganized and summarized the datasets in Table 2 to better present the results.

**Table 2. Overview and key parameters of flight missions**

| | Camera | Date | Mission Number | Flight Height (m) | Speed (m s$^{-1}$) | Area Covered (ha) | Satellite PDOP value | Ground Sampling Distance (cm px$^{-1}$) | Number of Images |
|---|---|---|---|---|---|---|---|---|---|
| **Before plowing** | DSLR camera (EOS) | 29.03.2018 | F1 | 45 | 3.4 | 3.75 | 1.3 | 0.6 | 323 |
| | | 05.04.2018 | F2_a | 45 | 3.4 | 3.26 | 1.2 | 0.6 | 360 |
| | | | F2_b | 45 | 3.4 | 3.26 | 1.2 | 0.6 | 362 |
| | Action camera (GoPro) | 29.03.2018 | F1_a | 45 | 3.4 | 11.33 | 1.3 | 3.1 | 134 |
| | | | F1_b | 45 | 3.4 | 13.27 | 1.2 | 3.1 | 155 |
| | | 30.03.2018 | F2 | 45 | 3.4 | 12.05 | 1.4 | 3.1 | 137 |
| **After plowing** | DSLR camera (EOS) | 06.04.2018 | F3_a | 35 | 3.0 | 0.85 | 1.3 | 0.5 | 129 |
| | | | F3_b | 35 | 3.0 | 0.8 | 1.2 | 0.5 | 107 |
| | | 06.04.2018 | F3_a | 20 | 2.6 | 3.23 | 1.2 | 1.3 | 182 |

| | | | | | | |
|---|---|---|---|---|---|---|---|
| Action camera (GoPro) | F3_b | 20 | 2.6 | 3.01 | 1.2 | 1.3 | 162 |

Note: Repeated flight missions were marked as F_a and F_b. The missions showed in the list were used parallel flight plan.

The accuracy results of each survey were presented using density plots (Fig. 4), showing the CP residuals on X, Y and Z directions. The units were given in both meters and pixels. The pixels were calculated based on GSD at corresponding flight
5    height.

*3) Line 3 page 6: Can you provide more information on the decision not to use a crossflight pattern or a single perpendicular strip as recommended by various authors?*

The UAV autonomy (flight duration) is an important factor to take into account. A crosshatch pattern could improve accuracy
10    by providing multi-angle images and increased overlap. However, it doubles the survey time and battery consumption, and may average out systematic bias in positioning. We therefore only used parallel flight lines in the main text. We also wanted to investigate the large-FOV (field of view) action camera, so as to find a better solution for accuracy/efficiency ratio.

For the surveys we implemented, we did flights using 'crosshatch' pattern. However, to simplify the manuscript we only used the 'parallel' flight pattern, which were listed in Table 2. We can also provide the results of 'crosshatch' missions, as a
15    comparison between the two flight patterns. We noticed that for the PPK direct georeferencing, a crosshatch pattern had slightly better performance on the altimetric accuracy.

[Figure]

**Figure. Distribution of CP residuals on X, Y and Z directions of surveys. Datasets: combined results of surveys using crosshatch pattern and parallel pattern, respectively.**

*4) Did your UAVs also record attitude parameters? In the manuscript, you should make clear why you used some observations of the external orientation parameters and others not.*

We apologize that this was not clear. Our cameras used a separate system without connection to IMU, so the images only contained positional information without attitude parameters. We have rephrased the method section to make it clear.

*5) You used Pix4D as a kind of black-box program. Which settings did you choose for camera calibration and the accuracy for geotagging information?*

We added details in 2.4.2 Point Cloud and DSM Generation. 'The horizontal and vertical accuracy were both set as 0.05 m. We kept the remaining settings as default as 3D maps template, i.e., full keypoints image scale, automatic targeted number of keypoints and standard calibration method.'

*6) What are the reasons for the artificial quadratic pattern in the DoD of the DSLR in Figure 6a (upper left picture) – this almost looks like a kind of systematic error. This pattern needs to be explained in the text.*

The artificial quadratic pattern of the DSLR camera were observed especially for the low-FOV camera. With a diagonal FOV ca. 43°, one tie point can be observed simultaneously by 7-12 images for EOS camera in our case. While we compared the output with the large-FOV camera GoPro and one single point can be captured by 60-80 images from wider angle. This had different effects on the BBA process, e.g., GoPro output had no such outline effects. Though the image overlap outlines shift in our case is ca. 2 cm, which can be eliminated by applying DoD threshold. We have added related discussion.

*7) Some results are not very clear to the reader. I recommend extending the results section with some more explanation to enhance readability.*

We have followed this suggestion and have rewritten the result section.

**Technical corrections:**

*1) Figures: Check consistency of font and readability of legends*

We have adjusted the size to improve the readability.

*2) There are some typos and grammar mistakes – I recommend English proofreading prior to publication.*

We have corrected these typos.

[revised manuscript text omitted]

~~Another feature of the low FOV DSLR camera was that the image overlap outlines clearly existed. It suggested that camera positions may have been over-constrained in the processing when we set the camera accuracy as 0.05 m. With increased camera accuracy (e.g., 0.1 m), the quadrate outline can be mitigated whilst the accuracy may degrade due to a weaken control of camera position. We explored how precision varied with different camera accuracy using DSLR dataset. The camera accuracies were set as 0.01 m, 0.03 m, 0.05 m, 0.1 m, 0.15 m, 0.2 m, 0.5 m and 1 m. The precisions were validated using CP residuals and corresponding MC extracted values (Fig. A3). The strong correlations indicated the precision of direct georeferencing is highly dependent on camera accuracy. Therefore, a prudent selection for camera accuracy is required. In our case, we maintained the rigorous constraint since the outline shifts were ca. 0.02 m, which can be eliminated by applying LoD threshold.~~This feature may explain the observed absence of a significant correlation between error and tie point density for the action camera (Table A3). We performed an additional analysis to verify this by only using half of the images (i.e., we removed one out of every two images) of the action camera dataset: the results showed that the error (based on sixteen control points) did not increase when using a substantially decreased tie point density.

It can be noticed that a correlation exists between tie point density and accuracy for the low FOV DSLR dataset. ~~Although the number of images was excluded in the multiple linear analyses, we found this factor to be strongly correlated to tie point density (Fig. 5aFig. 8). We inferred that denser tie points generally lead to robust positioning of tie points (Fig. A4). It implies that areas with relatively higher tie point density are of higher probability to be accurate and precise in the SfM output. 
[revised manuscript text omitted]